# DiffLight: A Partial Rewards Conditioned Diffusion Model for Traffic Signal Control with Missing Data

**Hanyang Chen**[1,2], **Yang Jiang**[1,2], **Shengnan Guo**[1,2,*] **Xiaowei Mao**[1,2],
**Youfang Lin**[1,2], **Huaiyu Wan**[1,2]
[1]School of Computer Science and Technology, Beijing Jiaotong University, China
[2]Beijing Key Laboratory of Traffic Data Analysis and Mining, Beijing, China
{hanyangchen, jiangyang, guoshn, maoxiaowei, yflin, hywan}@bjtu.edu.cn

## Abstract

The application of reinforcement learning in traffic signal control (TSC) has been extensively researched and yielded notable achievements. However, most existing works for TSC assume that traffic data from all surrounding intersections is fully and continuously available through sensors. In real-world applications, this assumption often fails due to sensor malfunctions or data loss, making TSC with missing data a critical challenge. To meet the needs of practical applications, we introduce DiffLight, a novel conditional diffusion model for TSC under data-missing scenarios in the offline setting. Specifically, we integrate two essential sub-tasks, *i.e.*, traffic data imputation and decision-making, by leveraging a Partial Rewards Conditioned Diffusion (PRCD) model to prevent missing rewards from interfering with the learning process. Meanwhile, to effectively capture the spatial-temporal dependencies among intersections, we design a Spatial-Temporal transFormer (STFormer) architecture. In addition, we propose a Diffusion Communication Mechanism (DCM) to promote better communication and control performance under data-missing scenarios. Extensive experiments on five datasets with various data-missing scenarios demonstrate that DiffLight is an effective controller to address TSC with missing data. The code of DiffLight is released at https://github.com/lokol5579/DiffLight-release.

## 1 Introduction

With the acceleration of urbanization, the surge in the number of vehicles in cities has led to increasingly severe traffic congestion and pollution problems [1]. Intersections, where traffic congestion often occurs, become a key in addressing these problems. To this end, solving the traffic signal control (TSC) problem is crucial for reducing traffic congestion in intersections by controlling traffic lights. Over the years, many approaches have been developed to tackle the TSC problem, which can be categorized into conventional approaches and reinforcement learning-based (RL-based) approaches. Conventional approaches, like Fixed-time [2], SCOOT [3] and SCATS [4], have been widely deployed in different cities. However, these approaches struggle to adapt to the inherent stochasticity and highly dynamic nature of real-time traffic conditions, limiting their effectiveness in responding to dynamic traffic demands.

Recently, reinforcement learning (RL) is introduced into TSC to enable adaptive traffic signal control [5, 6, 7, 8, 9, 10, 11, 12, 13]. Unlike conventional approaches, RL-based approaches for TSC deploy a learnable agent at each intersection, allowing traffic signals to be adjusted dynamically based on real-time traffic conditions. However, most existing RL-based approaches for TSC assume that traffic

---

*Corresponding author

data from all surrounding intersections is fully and continuously available through deployed sensors. In practice, this assumption is often unrealistic. Due to budget constraints, not all intersections can be equipped with sufficient sensors. Even if necessary sensors are deployed, malfunctions or errors could lead to incomplete data collection. Therefore, the research on TSC with missing data is more in line with the needs of actual scenarios but has not been studied sufficiently yet.

Furthermore, existing RL-based approaches for TSC can be categorized into two types, *i.e.*, online approaches and offline approaches. Most RL-based approaches for TSC rely on the online setting, interacting with the environment frequently. Specific to data-missing scenarios, MissLight [14] composed of the traffic data imputation stage and decision-making stage has been proposed in the online setting. However, frequent interaction with the real-world traffic environment is challenging and potentially unsafe, especially when dealing with incomplete data. As an alternative, training using offline traffic data with missing values offers a safer and more practical solution. Therefore, we focus on the offline setting in this paper. Similar to the online setting, traffic data imputation and decision-making for TSC with missing data are two sub-tasks we must confront in the offline setting.

Recently, diffusion models [15] have been introduced into offline RL due to their powerful generative ability [16, 17, 18, 19, 20]. These approaches frame sequential decision-making as conditional generative modeling and utilize the generative ability of the diffusion model to capture complex policy distribution in offline datasets to make better decisions. Additionally, in the context of TSC with missing data, traffic data imputation is equally critical. Inspired by existing works [21, 22, 23], we approach traffic data imputation as a conditional generative problem, similar to decision-making. Considering the similarity of the two sub-tasks, we propose to unify traffic data imputation and decision-making for TSC with missing data by utilizing the powerful generative ability of the diffusion model.

There are several challenges that must be addressed to integrate the two sub-tasks mentioned above effectively. Firstly, in RL-based approaches for TSC, rewards which are typically vehicle queue length, are critical for the performance of controllers. However, due to the absence of traffic data, only partial rewards are available. A straightforward solution might be to fill in the missing rewards with padded values, which could confuse the imputed rewards with the actual ones, leading to a negative impact on performance. Secondly, relying solely on traffic data of the local intersection makes it challenging to capture the dynamic and spatial-temporal dependencies in the traffic network for traffic data imputation and decision-making tasks. The complexity of traffic flow often requires a broader context, as traffic data from the local intersection may not adequately reflect the behaviors and interactions occurring across the entire network. The absence of traffic data from neighboring intersections may further exacerbate this issue, hindering the ability to capture these dependencies and potentially leading to a decline in performance.

To tackle these challenges, we introduce DiffLight, a novel conditional diffusion model for TSC with missing data. We propose a Partial Rewards Conditioned Diffusion (PRCD) model for both traffic data imputation and decision-making under data-missing scenarios to prevent missing rewards from interfering with the learning process. Meanwhile, to effectively capture the spatial-temporal dependencies among intersections, we design the noise model as a Spatial-Temporal transFormer (STFormer) architecture. In addition, we propose a Diffusion Communication Mechanism (DCM) to enable communication and promote the capture of spatio-temporal dependencies in the traffic network through the propagation of generated observations, facilitating better control in scenarios with missing data. Extensive experiments on five datasets with various data-missing scenarios are conducted to evaluate the effectiveness of DiffLight. The experimental results indicate that DiffLight is highly competitive for TSC with missing data.

## 2 Preliminaries

### 2.1 Partially Observable Markov Decision Process

We consider a partially observable Markov decision process (POMDP) in the offline setting, defined as a tuple $\langle \mathcal{S}, \mathcal{A}, \mathcal{P}, \mathcal{R}, \Omega, \mathcal{O}, \gamma \rangle$. $\mathcal{S}$ is the state space, and $s_t \in \mathcal{S}$ denotes the state at time $t$. $\mathcal{A}$ is the set of available actions and $a_t \in \mathcal{A}$ denotes the action of an agent at time $t$. The observation $o_t \in \Omega$ observed by the agent is part of the state $s_t$ and can be derived from the function $\mathcal{O}(s_t)$. $r_t = \mathcal{R}(s_t)$ is the immediate reward of an agent at time $t$. $\mathcal{P}$ and $\gamma$ denote the transition probability function

and the discount factor separately. The optimization objective is to learn a policy $\pi$ for agents to maximize the expected return $\mathbb{E}_{s_t,a_t}[R_t]$, where $R_t = \sum_t \gamma^t r_t$.

## 2.2 Traffic Signal Control with Missing Data

We formulate TSC with missing data as POMDP, and consider TSC in a traffic network with several intersections. Agents are deployed at each intersection of the traffic network. As illustrated in Figure 1, for a four-way intersection, there are twelve traffic movements from the intersection's entrance lane $l_{\text{in}}$ to the departure lane $l_{\text{out}}$, and four pairs of traffic movements without conflict comprise four traffic signal phases, *i.e.*, A, B, C, D. For example, the traffic signal phase of the intersection in Figure 1 is phase-A which involves movement-2 $= \{l_{\text{in}}^2 \rightarrow l_{\text{out}}^7, l_{\text{in}}^2 \rightarrow l_{\text{out}}^8, l_{\text{in}}^2 \rightarrow l_{\text{out}}^9\}$ and movement-8 $= \{l_{\text{in}}^8 \rightarrow l_{\text{out}}^1, l_{\text{in}}^8 \rightarrow l_{\text{out}}^2, l_{\text{in}}^8 \rightarrow l_{\text{out}}^3\}$.

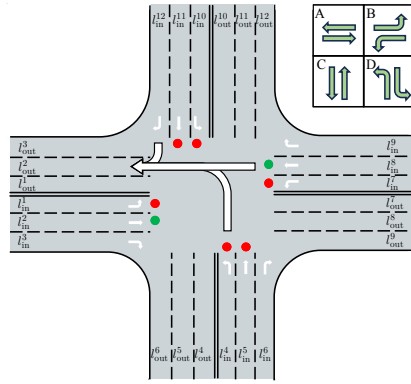

Figure 1: Illustration of a four-way intersection with 12 traffic movements and 4 traffic signal phases.

In this paper, the **observation** $o_t$ contains the number of vehicles $L_{\text{num}}$ and queue length $L_{\text{queue}}$ in every entrance lane of the local intersection. Available **actions** are four phases. The **immediate reward** $r_t$ is defined as the sum of the queue length $\sum_{l_{\text{in}}} L_{\text{queue}}$. Due to the lack or error of sensors resulting in missing traffic data, $o_t$ and $r_t$, which are derived from traffic data, could be missing in a particular missing pattern. We consider random missing and kriging missing patterns detailed in Appendix C.2. To simplify the problem, we assume that $o_t$ and $r_t$ in an intersection are missing simultaneously.

## 2.3 Diffusion Models for Reinforcement Learning

**Diffusion model.** Diffusion models [15, 24, 25], as a powerful generative model, provide a framework to model the data generative process as a discrete diffusion process. Diffusion models consist of two processes: the forward process and the reverse process. In this paper, the forward process is defined as $q(x^k|x^{k-1}) := \mathcal{N}(\sqrt{1 - \beta^k}x^{k-1}, \beta^k\mathbf{I})$ by the Markov process, where $\beta^k$ is the variance of the noise at timestep $k$. We adopt DDIM sampler [24] to sample in the reverse process in order to accelerate sampling. DDIM sampler is parameterized with $p_\theta(x^{k-1}|x^k, x^0) := \mathcal{N}(\mu_\theta(x^k, k), (\sigma^k)^2\mathbf{I})$, which can be optimized by a simplified surrogate loss,

$$\mathcal{L}(\theta) := \mathbb{E}_{k\sim\mathcal{U}(1,K),\epsilon\sim\mathcal{N}(\mathbf{0},\mathbf{I})}[\|\epsilon_\theta(x^k, k) - \epsilon\|^2]. \tag{1}$$

The reverse process begins by sampling an initial noise $x_K \sim \mathcal{N}(\mathbf{0}, \mathbf{I})$. The estimated mean of Gaussian is $\mu_\theta(x^k, k) = \sqrt{\bar{\alpha}^{k-1}}\hat{x}_0 + \sqrt{1 - \bar{\alpha}^{k-1}}\epsilon_\theta(x^k, k)$ where $\hat{x}_0 = \frac{1}{\sqrt{\bar{\alpha}^k}}(x^k - \sqrt{1 - \bar{\alpha}^k}\epsilon_\theta(x^k, k))$, $\alpha^k = 1 - \beta^k$, $\bar{\alpha}^k = \prod_{k=1}^{K} \alpha^k$ and $\epsilon_\theta(x^k, k)$ is a predictor used to estimate noise.

**Diffusing decision-making.** Recently, many diffusion-based approaches have been proposed to address decision-making problems in RL. Among existing works, Diffuser [16] chooses to diffuse on an observation-action trajectory with returns as the condition to generate actions directly. Decision Diffuser [16] focuses solely on diffusing the observation trajectory conditioned on returns. By avoiding direct diffusion over actions, Decision Diffuser enhances performance in scenarios with discrete actions. In this paper, considering the discrete nature of actions in TSC, we focus on introducing the approach diffusing on the observation trajectory $\tau$ sampled from offline dataset $\mathcal{D}$. We denote the $k$-step denoised output of the diffusion model as $x^k(\tau)$. The observation trajectory would be diffused to generate $o_{t+1}$. However, only diffusing on observation trajectory is not enough to make decisions. An inverse dynamics model $f_\phi$ is adapted to generate the action $a_t$ that makes the observation transit from $o_t$ to $o_{t+1}$,

$$a_t := f_\phi(o_t, o_{t+1}). \tag{2}$$

**Classifier-free guidance.**   Classifier-free guidance [26] aims to learn the conditional distribution $q(x(\tau)|y(\tau))$. It learns both a conditional $\epsilon_\theta(x^k(\tau), y(\tau), k)$ and an unconditional $\epsilon_\theta(x^k(\tau), \phi, k)$ for the noise. Then, the perturbed noise $\epsilon_\theta(x^k(\tau), \phi, k) + \omega(\epsilon_\theta(x^k(\tau), y(\tau), k) - \epsilon_\theta(x^k(\tau), \phi, k))$ can be used to generate samples, where $\omega$ is the guidance scale.

## 3   Methodology

In order to effectively unify traffic data imputation and decision-making for TSC with missing data, we consider both of them as a conditional generative modeling problem via diffusion models,

$$\max_\theta \mathbb{E}_{\tau \sim \mathcal{D}}[\log p_\theta(x^0(\tau)|y(\tau))], \tag{3}$$

where $p_\theta$ is a learnable model distribution to estimate the conditional data distribution of trajectory $x^0(\tau)$, conditioned on $y(\tau)$. We construct our generative model according to the conditional diffusion process,

$$q(x^k(\tau)|x^{k-1}(\tau)), \qquad p_\theta(x^{k-1}(\tau)|x^k(\tau), x^0(\tau), y(\tau)), \tag{4}$$

with conditions as,

$$y(\tau) := [r(\tau), y'(\tau)], \qquad y'(\tau) := [\tau'_{\text{obs}}, \tau'_{\text{nei}}]. \tag{5}$$

To facilitate a better understanding of the symbol definitions, we categorize the trajectory segments into three types. The **observed** trajectory consists of data collected by sensors up to time $t$. The **missing** trajectory includes data that have not been collected during this period. The **observable** trajectory encompasses both the observed trajectory and the data that could potentially be collected in the future. In this context, $r(\tau)$ is the observable reward trajectory, and $\tau'_{\text{obs}}$ is the observed part of trajectory $\tau$ from the local intersection. $\tau'_{\text{nei}} = \cup_N f_{\text{nei}}(\tau^i)$ is the observed observations from neighboring intersections, where $\tau^i$ denotes the observation trajectory of the neighboring intersection $i$, $N$ is total number of neighboring intersections, $f_{\text{nei}}(\cdot)$ represents the observed observations from entrance lanes of all neighboring intersections that feed into the entrance lanes of the local intersection, as shown in Figure 1. Due to discrete and high-frequent actions in TSC, we choose to diffuse solely on observations and utilize the inverse dynamic model to generate actions, which demonstrates a better performance proven in Appendix F.1 and F.6. We define the observation trajectory $\tau$ under data-missing scenarios as,

$$\tau := [\hat{o}_{t-C+1}, o_{t-C+2}, \cdots, \hat{o}_t, \hat{o}_{t+1}, \cdots, \hat{o}_{t+H}], \tag{6}$$

where $o_t$ is the observation collected by sensors, $\hat{o}_t$ is the uncollected observation up to time $t$ or a potentially collectible observation in the future, $C$ is the length of historical observations and $H$ is the horizon of future observations. It should be noted that we generate $H$-step future observations to enable effective long-horizon planning [16].

Figure 2 shows the overview of DiffLight, which consists of the Partial Rewards Conditioned Diffusion (PRCD), a noise model with a Spatial-Temporal transFormer structure (STFormer), and the Diffusion Communication Mechanism (DCM). We introduce each of them in the following sections.

### 3.1   Partial Rewards Conditioned Diffusion

Under data-missing scenarios, sensors deployed to collect traffic data for rewards may malfunction or be lacking. The absence of partial rewards makes it challenging to calculate returns. Therefore, we adopt partial rewards as the condition instead of returns [16, 17, 19], which can be expressed as,

$$r(\tau) := [\tilde{r}_{t-C+1}, r_{t-C+2}, \cdots, \tilde{r}_t, r_{t+1}, \cdots, r_{t+H}], \tag{7}$$

where $r_t$ is the collected reward or a potentially collectible reward in the future, and $\tilde{r}_t$ is the uncollected reward.

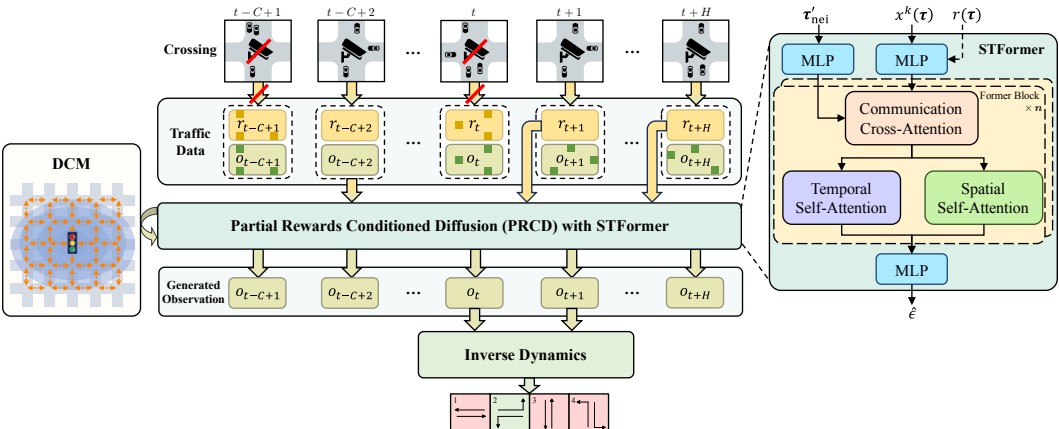

Figure 2: An overview of DiffLight. We demonstrate the signal control process of an intersection in random missing. Traffic data is collected by sensors to derive rewards and observations. Missing rewards and observations are masked. Only the observed part of the observation trajectory and observable rewards from the local intersection, and observation trajectories from neighboring intersections would be input into PRCD with STFormer. In the inference process, DCM would work with STFormer to generate observations. The inverse dynamics model is used to generate actions to control the traffic signal.

There are two ways to handle the missing part of rewards: conditioning on the rewards with padded values or conditioning on the partial observable rewards directly. Padding a specific value in the missing part is a feasible way. However, padded values could confuse the imputed rewards with the actual ones, leading to a negative impact on performance proven in Section 4.3. For better decision-making, we choose to condition on only partial observable rewards. We assume that the observable part and the missing part of the trajectory are collected by real sensors and virtual sensors separately, and the distribution of traffic data collected by two kinds of sensors is independent. In this case, the distribution in Equation 3 can be factorized as,

$$p_\theta(x^0(\boldsymbol{\tau})|y(\boldsymbol{\tau})) = p_\theta(x^0(\boldsymbol{\tau}_{\text{obs}})|r(\boldsymbol{\tau}), y'(\boldsymbol{\tau})) \cdot p_\theta(x^0(\boldsymbol{\tau}_{\text{mis}})|y'(\boldsymbol{\tau})), \tag{8}$$

where $\boldsymbol{\tau}_{\text{obs}}$ is the observable part of the trajectory $\boldsymbol{\tau}$ from the local intersection, and $\boldsymbol{\tau}_{\text{mis}}$ is the missing part. We parameterize Equation 8 as the same form of locally conditioned diffusion [27, 28], propose partial rewards conditioned diffusion (PRCD) with classifier-free guidance [26] and introduce it into TSC with missing data. Given condition set $\mathbf{c} = \{\phi, r(\boldsymbol{\tau})\}$ and binary non-overlapping mask set $\mathbf{m} = \{m_{\text{mis}}, m_{\text{obs}}\}$, PRCD assigns partial rewards to corresponding observation sub-trajectory masked by $\mathbf{m}$, which can be formulated as,

$$\begin{aligned}\hat{\epsilon}_\theta(x^k(\boldsymbol{\tau}), k, y'(\boldsymbol{\tau}), \mathbf{c}, \mathbf{m}) :=& m_{\text{obs}} \odot \hat{\epsilon}_\theta(x^k(\boldsymbol{\tau}), k, y'(\boldsymbol{\tau}), r(\boldsymbol{\tau})) + \\ & m_{\text{mis}} \odot \hat{\epsilon}_\theta(x^k(\boldsymbol{\tau}), k, y'(\boldsymbol{\tau}), \phi),\end{aligned} \tag{9}$$

where $\hat{\epsilon}_\theta(x^k(\boldsymbol{\tau}), k, y'(\boldsymbol{\tau}), c_i)$ could be expressed using classifier-free guidance,

$$\begin{aligned}\hat{\epsilon}_\theta(x^k(\boldsymbol{\tau}), k, y'(\boldsymbol{\tau}), c_i) :=& \epsilon_\theta(x^k(\boldsymbol{\tau}), k, y'(\boldsymbol{\tau}), \phi) + \\ & \omega(\epsilon_\theta(x^k(\boldsymbol{\tau}), k, y'(\boldsymbol{\tau}), c_i) - \epsilon_\theta(x^k(\boldsymbol{\tau}), k, y'(\boldsymbol{\tau}), \phi)),\end{aligned} \tag{10}$$

where $c_i \in \mathbf{c}$. We provide a derivation for the feasibility of PRCD in Appendix E.

## 3.2 Diffusing with Spatial-Temporal Transformer

The noise model with a U-Net structure is widely applied in image generation [15, 24, 29, 30], control [16, 17, 19] and other fields. However, it is hard to be applied to capture the spatial-temporal

dependencies in TSC. The emergence of Transformer [31] and its applications on spatial-temporal modeling [32, 33, 34, 35] provide a promising solution to deal with it. In this section, we design a Spatial-Temporal transFormer (STFormer) structure to effectively model the spatial-temporal dependencies in TSC, which includes a data embedding layer, stacked $L$ spatial-temporal encoder layers, and an output layer. Data embedding layer embeds different inputs into embeddings, including diffusion timestep, trajectory timestep, rewards, trajectory of the local intersection, and neighboring intersections. Spatial-Temporal Encoder layer (STE) is composed of Communication Cross-Attention module (CCA), Spatial Self-Attention module (SSA), and Temporal Self-Attention module (TSA). CCA is designed to capture the spatial-temporal dependencies between the local intersection and neighboring intersections. SSA and TSA are designed to capture the spatial dependencies and temporal dependencies at the local intersection separately. Output layer is used to convert the output of STE into the noise we desire to predict. We detail the structure of STFormer in Appendix B.2.

### 3.3 Diffusion Communication Mechanism

Observations of neighboring intersections are crucial for TSC with missing data [14]. However, due to the possible absence of observations from neighboring intersections, the traffic signal could be controlled ineffectively. For instance, we assume an extreme situation where there is no available observation in both the local intersection and neighboring intersections. In this case, observation trajectories of intersections are all masked by noise, leading to difficulty in decision-making at the local intersection. Therefore, we propose a Diffusion Communication Mechanism (DCM) to disseminate observation information generated by the noise model in the reverse process among intersections. Formally, we formulate DCM as,

$$
\boldsymbol{\tau}'_{\text{nei}} = \begin{cases} \cup_N f_{\text{nei}}(\boldsymbol{\tau}^i), & k = K, \\ \cup_N f_{\text{nei}}(\frac{1}{\sqrt{\bar{\alpha}^k}}(x^k(\boldsymbol{\tau}^i) - \sqrt{1 - \bar{\alpha}^k}\hat{\epsilon}_\theta(x^k(\boldsymbol{\tau}), k, y'(\boldsymbol{\tau}), \mathbf{c}, \mathbf{m}))), & k < K. \end{cases} \tag{11}
$$

The reverse process begins by inputting original observations of neighboring intersections with missing data. During diffusing, we predict $\hat{x}^0(\boldsymbol{\tau}^i)$, which is the same in Section 2.3. With the help of DCM, generated observations of neighboring intersections could be spread from neighboring intersections for better decisions of the agent at the local intersection. Note that we train our model with ground-truth values collected from neighboring intersections and only use DCM in the inference process.

### 3.4 Training and Inference of DiffLight

**Training process.** Given an offline dataset $\mathcal{D}$ which consists of observation trajectories, rewards and actions, we train the reverse process $p_\theta$ parameterized through the noise model $\epsilon_\theta$, and the inverse dynamics model $f_\phi$ in DiffLight with the following loss,

$$
\begin{aligned}
\mathcal{L}(\theta) :=& \mathbb{E}_{(o,a,o')\in\mathcal{D}}[\|a - f_\phi(o, o')\|^2 \cdot \mathbb{1}(o, o')] + \\
& \mathbb{E}_{k,\epsilon,\beta\sim\text{Bern}(p)}[\|\hat{\epsilon}_\theta(x^k(\boldsymbol{\tau}), k, y'(\boldsymbol{\tau}), (1-\beta)r(\boldsymbol{\tau}) + \beta\phi, \mathbf{m}) - \epsilon\|^2].
\end{aligned} \tag{12}
$$

For each trajectory $\boldsymbol{\tau}$, we sample noise $\epsilon \sim \mathcal{N}(\mathbf{0}, \mathbf{I})$ and a diffusion timestep $k \sim \mathcal{U}(1, K)$, construct a masked noise array of observations $x^k(\boldsymbol{\tau})$ with $\boldsymbol{\tau}'_{\text{obs}}$, and predict the noise $\hat{\epsilon}_\theta(x^k(\boldsymbol{\tau}), k, y'(\boldsymbol{\tau}), r(\boldsymbol{\tau}), \mathbf{m})$. Missing values in condition $\boldsymbol{\tau}'_{\text{nei}}$ are padded with zeros. It should be noted that we ignore the rewards condition $r(\boldsymbol{\tau})$ with probability $p$ and the inverse dynamics is trained with individual transitions without missing observation $o$ or $o'$. For the training process of DiffLight, due to the inaccessibility of the ground-truth of missing data, we consider it self-supervised learning. In random missing, given a trajectory $\boldsymbol{\tau}$ and conditions, we can separate the observed part into two parts and set one of them to miss. In kriging missing, we randomly mask the whole trajectories of one observed intersection. Then, we can train the noise model $\hat{\epsilon}_\theta$ by solving Equation 12.

**Inference process.** DiffLight is deployed to every intersection in the traffic network in the inference process. Given rewards $r(\boldsymbol{\tau})$, a $C$-step observed trajectory of local intersection $\boldsymbol{\tau}'_{\text{obs}}$ with missing

data and trajectories of neighboring intersections $\tau'_{\mathrm{nei}}$, the agent can impute the missing observations of local intersection, predict the observations in the future and generate next action with Equation 2, 9 and 10. In order to sample a high reward trajectory, the rewards from time $t+1$ to $t+H$ are considered in $r(\tau)$, which is set to 1.

# 4 Experiments

## 4.1 Experimental Setup

**Experiment Settings** We conduct our experiments on CityFlow [36], a traffic simulator widely used in various RL-based methods. Similar to existing work in [13], we set the phase number as four and the minimum action duration as 15 seconds.

**Datasets** The datasets consist of two parts: offline datasets with missing data and real-world traffic flow datasets with traffic networks. We apply five real-world traffic flow datasets [9, 37] for comparison, including Hangzhou$_1$ ($\mathcal{D}^1_{\mathrm{HZ}}$), Hangzhou$_2$ ($\mathcal{D}^2_{\mathrm{HZ}}$), Jinan$_1$ ($\mathcal{D}^1_{\mathrm{JN}}$), Jinan$_2$ ($\mathcal{D}^2_{\mathrm{JN}}$) and Jinan$_3$ ($\mathcal{D}^3_{\mathrm{JN}}$). Corresponding offline datasets are composed of training trajectories of three online methods. We detail the datasets in Appendix C.1. To simulate data-missing scenarios, we adopt masks of different missing rates in different missing patterns, including random missing (RM) and kriging missing (KM), to mask observations and rewards. We describe the details of two patterns and missing rates in Appendix C.2.

**Evaluation Metrics** We use the average travel time (ATT) as the main metric for evaluation, which is widely used to evaluate the performance of TSC. It calculates the average time all the vehicles spend between entering and leaving the traffic network during simulation, which is formulated as,

$$\mathrm{ATT} = \frac{1}{N} \sum_{i=1}^{N} \left( t_i^l - t_i^e \right),\qquad(13)$$

where $N$ is the total number of vehicles entering the road network, $t_i^e$ and $t_i^l$ are the entering time and leaving time for the $i$-th vehicle respectively. The lower ATT indicates a better control performance.

**Compared Methods** We compare our method with Behavior Cloning (BC) [38] and offline approaches, including CQL [39], TD3+BC [40], Decision Transformer (DT) [41], Diffuser [16], Decision Diffuser (DD) [17]. Similar to the existing work in [14], we adopt store-and-forward method (SFM) [42], a rule-based method that has generally more stable performances, to impute missing observations and rewards for these approaches. We detail these approaches and SFM in Appendix D.

## 4.2 Performance under Data-Missing Scenarios

We train and test our method on all five datasets with different missing patterns and missing rates, and compare our method with all baselines. DiffLight performs the best on most of the datasets. We analyze experiments of different missing patterns below.

**Random missing.** In random missing, we can see that DiffLight achieves optimal or sub-optimal performance compared with baselines in all datasets in Table 1. Diffusion-based approaches, including Diffuser and DD, show a better performance than other baselines in most datasets. These diffusion-based approaches utilize a noise model to predict noise and generate actions or observations, which helps mitigate the disturbance caused by imputed observations and rewards during the diffusion process. Compared with diffusion-based approaches, the performance of other baselines without the diffusion process is disturbed by imputed observations and rewards more seriously.

Table 1: Comparing ATT for DiffLight and baselines in random missing. We report the mean and the standard error for three trials.

| Dataset | Rate | BC | CQL | TD3+BC | DT | Diffuser | DD | DiffLight |
|---|---|---|---|---|---|---|---|---|
| $\mathcal{D}_{HZ}^1$ | 10% | 349.59 | 363.5 | 337.54 | 300.64 | 290.66 | 289.38 | **286.17**±0.87 |
| | 30% | 350.08 | 368.53 | 338.21 | 315.64 | 302.39 | 298.67 | **292.81**±0.66 |
| | 50% | 357.13 | 383.67 | 343.23 | 343.96 | 313.68 | 422.5 | **304.71**±2.12 |
| $\mathcal{D}_{HZ}^2$ | 10% | 382.45 | 353.23 | 370.16 | 347.25 | 346.82 | 347.77 | **327.13**±1.43 |
| | 30% | 388.73 | 352.55 | 376.06 | 360.59 | 366.24 | 364.6 | **330.68**±2.63 |
| | 50% | 387.77 | 367.38 | 375.32 | 377.79 | 398.23 | 395.61 | **333.90**±2.67 |
| $\mathcal{D}_{JN}^1$ | 10% | 320.6 | 299.07 | 315.54 | 308.78 | 272.51 | **260.76** | 272.18±0.93 |
| | 30% | 328.97 | 310.8 | 326.37 | 377.39 | 295.09 | 300.49 | **279.10**±2.10 |
| | 50% | 355.47 | 322.25 | 351.2 | 439.89 | 324.75 | 517.99 | **290.02**±2.18 |
| $\mathcal{D}_{JN}^2$ | 10% | 288.42 | 305.86 | 322.44 | 259.3 | 255.12 | **245.85** | 247.17±1.38 |
| | 30% | 297.26 | 308.13 | 330.43 | 263.24 | 271.53 | 256.16 | **254.87**±0.69 |
| | 50% | 299.44 | 320.17 | 334.78 | 278.22 | 302.28 | 275.2 | **268.29**±0.90 |
| $\mathcal{D}_{JN}^3$ | 10% | 301.35 | 291.26 | 281.75 | 257.66 | 246.90 | **242.56** | 246.65±0.94 |
| | 30% | 315.03 | 295.61 | 283.24 | 312.56 | 258.83 | 256.95 | **254.55**±0.35 |
| | 50% | 326.55 | 301.1 | 292.98 | 382.93 | 272.36 | 351.92 | **265.76**±0.01 |

**Kriging missing.** In kriging missing, DiffLight shows the best performance in most datasets in Table 2. Unlike the results of random missing, DD does not perform well in kriging missing compared with other baselines. Since DD must impute missing observations with the SFM model first, generate future observations with the diffusion model, and then use the inverse dynamics to generate actions, which leads to serious error accumulation. While other baselines generate actions directly, requiring only roughly imputed observations and rewards.

Table 2: Comparing ATT for DiffLight and baselines in kriging missing. We report the mean and the standard error for three trials.

| Dataset | Rate | BC | CQL | TD3+BC | DT | Diffuser | DD | DiffLight |
|---|---|---|---|---|---|---|---|---|
| $\mathcal{D}_{HZ}^1$ | 6.25% | 338.33 | 317.69 | 332.80 | 300.78 | 302.99 | 395.54 | **294.18**±3.36 |
| | 12.50% | 346.83 | 317.94 | 332.43 | 310.37 | 305.93 | 483.47 | **294.11**±4.34 |
| | 18.75% | 350.08 | 319.18 | 333.24 | 306.35 | 307.22 | 572.56 | **300.31**±0.31 |
| | 25.00% | 354.86 | 328.83 | 341.89 | 381.94 | 328.79 | 836.46 | **302.16**±1.23 |
| $\mathcal{D}_{HZ}^2$ | 6.25% | 380.18 | 354.08 | 374.04 | 347.53 | 363.69 | 370.80 | **330.40**±0.11 |
| | 12.50% | 375.93 | 361.52 | 374.66 | 363.5 | 378.51 | 424.99 | **319.11**±7.19 |
| | 18.75% | 380.74 | 362.82 | 376.48 | 374.69 | 413.48 | 435.13 | **327.61**±9.68 |
| | 25.00% | 413.46 | 418.97 | 390.75 | 492.56 | 378.54 | 590.69 | **351.21**±9.86 |
| $\mathcal{D}_{JN}^1$ | 8.33% | 319.85 | 302.35 | 317.17 | 306.52 | 332.44 | 595.34 | **280.75**±0.11 |
| | 16.67% | 339.19 | 343.16 | 349.72 | 380.97 | 349.74 | 643.48 | **306.06**±14.89 |
| | 25.00% | 392.91 | 398.66 | 391.32 | 432.56 | 410.5 | 995.99 | **329.67**±16.04 |
| $\mathcal{D}_{JN}^2$ | 8.33% | 287.29 | 306.94 | 319.4 | 261.98 | 259.51 | 460.22 | **254.13**±0.35 |
| | 16.67% | 299.41 | 314.43 | 321.88 | **267.67** | 270.15 | 731.49 | 272.76±1.42 |
| | 25.00% | 314.63 | 359.33 | 323.65 | 295.59 | **295.21** | 1049.19 | 325.20±26.63 |
| $\mathcal{D}_{JN}^3$ | 8.33% | 310.44 | 287.25 | 282.46 | 368.2 | 267.64 | 324.42 | **249.48**±0.16 |
| | 16.67% | 327.7 | 311.89 | 295.07 | 322.96 | 294.27 | 399.67 | **274.13**±2.50 |
| | 25.00% | 381.37 | 337.33 | 312.44 | 494.04 | **292.26** | 409.76 | 342.07±16.11 |

We provide the overall performance of DiffLight and baselines without missing data as well, which is detailed in Appendix F.1. In addition, we provide further experiments on the influence of unobserved locations of intersections in Appendix F.2, the limit of missing rates in Appendix F.3, and the scalability of the approach in Appendix F.4.

### 4.3 Ablation Study

We further evaluate the effectiveness of different parts in DiffLight with the following variants. (1) U-Net: this variant replaces STFormer with U-Net as the noise model and missing rewards input are zero-padded. (2) STFormer: this variant uses STFormer as the noise model and keeps missing rewards zero-padded. (3) STFormer+PRCD: this is equal to DiffLight which uses STFormer as the noise model and is conditioned on partial rewards. It should be noted that DCM is adopted in both STFormer and STFormer+PRCD and all missing observations would not be imputed by the SFM model but are masked with Gaussian noise in order to be inpainted in the reverse process in ablation experiments.

Table 3: Ablation study on Hangzhou$_1$ and Jinan$_1$.

| Dataset | Pattern and Rate | U-Net | STFormer | STFormer+PRCD |
|---|---|---|---|---|
| $\mathcal{D}_{HZ}^1$ | RM (50%) | 668.38±65.54 | 350.60±9.88 | **304.71**±2.12 |
| | KM (25%) | 363.80±44.65 | 318.21±5.84 | **302.16**±1.23 |
| $\mathcal{D}_{JN}^1$ | RM (50%) | 509.64±41.15 | 374.41±3.61 | **290.02**±2.18 |
| | KM (25%) | 454.90±65.10 | 374.23±69.90 | **329.67**±16.04 |

Table 3 shows the comparison of these variants on Hangzhou$_1$ and Jinan$_1$ with random missing and kriging missing. Based on the results, we can find that STFormer which takes spatial-temporal dependencies into consideration leads to a great performance improvement over U-Net. It shows that capturing spatial-temporal dependencies is important in TSC. STFormer+PRCD performs better than STFormer, indicating that padding values in rewards could be confused with the ground-truth rewards and only conditioning on partial rewards could have a better performance. Due to the space limitation, we provide further experiments on DCM in Appendix F.5 and the inverse dynamics in Appendix F.6.

### 4.4 Model Generalization

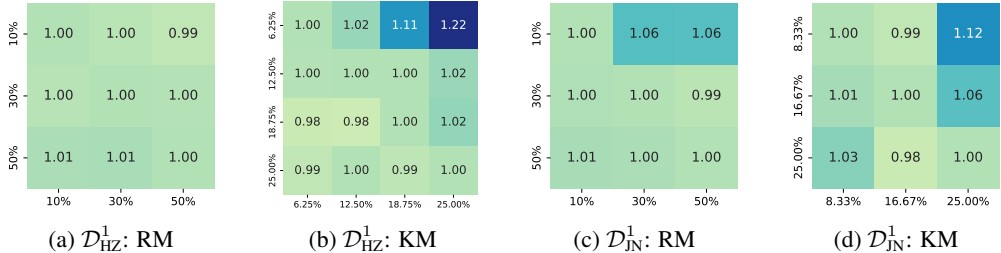

(a) $\mathcal{D}_{HZ}^1$: RM          (b) $\mathcal{D}_{HZ}^1$: KM          (c) $\mathcal{D}_{JN}^1$: RM          (d) $\mathcal{D}_{JN}^1$: KM

Figure 3: The relative generalization performance of DiffLight in different missing rates. The x-axis is the missing rate during testing and the y-axis is the missing rate during training. The formula used to calculate the relative generalization performance is depicted below.

We evaluate the generalization performance of DiffLight on Hangzhou$_1$ and Jinan$_1$ with different missing rates. We train our method in a specific missing rate and test it on the same dataset with other missing rates. To better compare the generalization performance among models trained in different missing rates, we formulate the relative generalization performance as,

$$P_r = P_g/P_o, \tag{14}$$

where $P_r$ is the relative generalization performance of the current missing rate, $P_g$ is the performance of the model trained in other missing rates, and $P_o$ is the performance of the model trained in the current missing rate. The results in Figure 3 demonstrate that the generalization performance of DiffLight is excellent in most situations. If we train DiffLight in a high missing rate and test it in a lower missing rate, the performance of DiffLight remains stable. In contrast, if we train DiffLight in a low missing rate and test it in a higher missing rate, the performance of DiffLight would decrease

slightly. As data with a higher missing rate has more complex missing situations, making it difficult for models to handle these situations.

## 5 Related Work

**Traffic Signal Control**  TSC approaches can be categorized into conventional and RL-based methods. Conventional approaches like Fixed-time [2], SCOOT [3] and SCATS [4] have been widely deployed in different cities. In recent years, RL-based approaches for TSC get more attention. DQN algorithm is introduced into TSC in [5, 6, 7] for dynamic real-time control. Attention mechanisms are applied to promote inter-agent communication [9] and build universal models [10]. Max-pressure [8] and advanced-MP [13] are proposed to promote the performance of existing methods. TSC with missing data is considered with the help of the imputation model in the online setting. However, there is no existing work to solve this problem in the offline setting.

**Diffusion-based Reinforcement Learning**  There are various works applying the diffusion model to offline RL recently. Diffusion and deep q-learning are combined [18], demonstrating the potential of diffusion in RL. The state-action trajectory is encoded in latent space [43], enhancing credit assignment and reward propagation. In addition to methods relying on TD-learning, Diffuser [16] and Decision Diffuser [17] are proposed as planners to generate the trajectory with a conditional diffusion model. However, they are all studied under scenarios without missing data, while we model TSC with missing data.

**Traffic Data Imputation**  With the development of deep learning, RNN-based methods [44, 45, 46] show good performance for traffic data imputation. In subsequent studies, diffusion models are utilized to learn the complex distribution in traffic data [21, 22]. For TSC, store-and-forward method (SFM) [42] is proven to have a more stable performance than neural networks [14]. In this paper, we adopt SFM to impute observations and rewards for baselines.

## 6 Conclusion and Limitation

**Conclusion**  In this paper, we introduce DiffLight, a novel conditional diffusion model designed for TSC in scenarios with missing data. Our approach centers on the Partial Rewards Conditioned Diffusion (PRCD) model, which addresses both traffic data imputation and decision-making in the presence of incomplete data. This model helps prevent missing rewards from disrupting the learning process. We address the challenge of capturing spatial-temporal dependencies across intersections by designing a Spatial-Temporal transFormer (STFormer) architecture as the noise model. Additionally, to enhance communication and control performance, we propose a Diffusion Communication Mechanism (DCM) that facilitates the propagation of generated observations. We conduct extensive experiments on different datasets and settings to demonstrate that DiffLight is an effective controller to address TSC with missing data.

**Limitation**  In this work, we only consider two missing patterns: random missing and kriging missing. While in the real world, the missing pattern in the traffic network is more complex. Meanwhile, we just adopt SFM which is similar to k-nearest neighbor (KNN) to impute the traffic data for baselines. In addition, our approach is conditioned on partial rewards instead of returns which could lead to the short-sightedness of agents. Future work could explore the influence on performance in more different missing patterns even mixed missing patterns, adopt more different imputation methods, and find out a more far-sighted method to control the traffic signals under data-missing scenarios.

## Acknowledgments

This work was supported by the National Natural Science Foundation of China (No. 62372031).

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

# A   Broad Impacts

Our proposed method demonstrates effective abilities in TSC with missing data. It can handle different missing patterns and different missing rates when controlling traffic signals. Even in an intersection where there are no observed neighboring intersections, DiffLight can perform competitively against baselines with SFM. Moreover, the fast inference speed with fewer sampling steps and stable performance indicates that DiffLight can be deployed to achieve real-time control in the real world. However, a potential negative impact of this work is that bad decisions could lead to a collapse in the traffic network.

# B   The Details of DiffLight

## B.1   Hyperparameters of DiffLight

In this section, we describe the details of hyperparameters,

- We set the batch size as 64 and each sample contains the trajectory of the whole intersections in the traffic network. We train our model using Adam optimizer [47] with $2e^{-4}$ learning rate for $1.5e^5$ train steps.
- We train DiffLight on NVIDIA GeForce RTX A5000 for around 15 hours and test it on the same GPU.
- We choose the probability $p$ of removing the condition information to be 0.25 and guidance scale $\alpha = 1.2$.
- In DiffLight, we choose the length of historical observations $C = 5$ and the planning horizon of observation trajectory $H = 3$.
- We use $K = 100$ for diffusion steps.

## B.2   Structure of STFormer

STFormer is composed of a data embedding layer, stacked $L$ spatial-temporal encoder layers, and an output layer. We detail each of them as follows.

**Data embedding layer.**   Different inputs are embedded into embeddings $\boldsymbol{e}$ with the same dimension $D$ by the data embedding layer which consist of separate MLPs $f_{\mathrm{MLP}}(\cdot)$,

$$
\begin{aligned}
\boldsymbol{e}_{\mathrm{dt}} &:= f_{\mathrm{MLP}}(k), \quad \boldsymbol{e}_{\mathrm{tt}} := f_{\mathrm{MLP}}(t_{0:T-1}), \quad \boldsymbol{e}_{\mathrm{r}} := f_{\mathrm{MLP}}(R(\boldsymbol{\tau})), \\
\boldsymbol{e}_{\mathrm{ctr}} &:= f_{\mathrm{MLP}}(\mathrm{x}^k(\boldsymbol{\tau})), \quad \boldsymbol{e}_{\mathrm{ntr}} := f_{\mathrm{MLP}}(\boldsymbol{\tau}_{\mathrm{nei}})
\end{aligned}
\tag{15}
$$

where $t_{0:T-1}$ is the timestep of trajectory, $\boldsymbol{e}_{\mathrm{dt}}$, $\boldsymbol{e}_{\mathrm{tt}}$, $\boldsymbol{e}_{\mathrm{r}}$, $\boldsymbol{e}_{\mathrm{ctr}}$ and $\boldsymbol{e}_{\mathrm{ntr}}$ represent the embedding of diffusion timestep, trajectory timestep, rewards, trajectory of local intersection and partial trajectory of neighboring intersection separately.

**Spatial-temporal encoder layer.**   The spatial-temporal encoder layer, abbreviated as STE, is composed of Communication Cross-Attention module, Spatial Self-Attention module and Temporal Self-Attention module. We adopt the vanilla attention operator [31] in modules, represented as $f_{\mathrm{Att}}(Q, K, V)$. The following slice notations are used to formulate attention modules. For the embedding of local intersection's trajectory $\boldsymbol{e}_{\mathrm{ctr}} \in \mathbb{R}^{T \times L \times D}$ where $L$ is the number of entrance lanes, the $t$ slice is $\boldsymbol{e}_{\mathrm{ctr}}^{t::} \in \mathbb{R}^{L \times D}$ and the $l$ slice is $\boldsymbol{e}_{\mathrm{ctr}}^{:l:} \in \mathbb{R}^{T \times D}$. For the embedding of neighboring intersections' partial trajectories $\boldsymbol{e}_{\mathrm{ntr}} \in \mathbb{R}^{L \times (T \cdot L') \times D}$ where $L'$ is the number of neighboring intersections' entrance lanes taken into consideration, the $l$ slice is $\boldsymbol{e}_{\mathrm{ntr}}^{l::} \in \mathbb{R}^{(T \cdot L') \times D}$.

The Communication Cross-Attention module, abbreviated as CCA, is designed to capture the spatial-temporal dependencies between the local intersection and neighboring intersections. As illustrated in Figure 1, $\boldsymbol{e}_{\mathrm{ntr}}^{l::}$ contains information of entrance lanes from neighboring intersections feeding into lane $l$. This module can be formulated as,

$$
f_{\mathrm{CCA}}(\boldsymbol{e}_{\mathrm{ctr}}^{:l:}, \boldsymbol{e}_{\mathrm{ntr}}^{l::}) := f_{\mathrm{Att}}(\boldsymbol{e}_{\mathrm{ctr}}^{:l:}, \boldsymbol{e}_{\mathrm{ntr}}^{l::}, \boldsymbol{e}_{\mathrm{ntr}}^{l::})
\tag{16}
$$

$$
\boldsymbol{e}_{\mathrm{ctr}}'^{:l:} = f_{\mathrm{CCA}}(\boldsymbol{e}_{\mathrm{ctr}}^{:l:}, \boldsymbol{e}_{\mathrm{ntr}}^{l::}) + \boldsymbol{e}_{\mathrm{ctr}}^{:l:}
\tag{17}
$$

The Spatial Self-Attention module, abbreviated as SSA, and Temporal Self-Attention module, abbreviated as TSA, are designed to capture the spatial dependencies and temporal dependencies separately in the local intersection, which can be formulated as,

$$f_{\text{SSA}}(\boldsymbol{e}_{\text{ctr}}'^{h::}) := f_{\text{Att}}(\boldsymbol{e}_{\text{ctr}}'^{t::}, \boldsymbol{e}_{\text{ctr}}'^{t::}, \boldsymbol{e}_{\text{ctr}}'^{t::}), \quad f_{\text{TSA}}(\boldsymbol{e}_{\text{ctr}}'^{:l:}) := f_{\text{Att}}(\boldsymbol{e}_{\text{ctr}}'^{:l:}, \boldsymbol{e}_{\text{ctr}}'^{:l:}, \boldsymbol{e}_{\text{ctr}}'^{:l:}) \tag{18}$$

Therefore, the spatial-temporal encoder layer can be expressed as,

$$f_{\text{STE}}(\boldsymbol{e}_{\text{ctr}}, \boldsymbol{e}_{\text{ntr}}) := f_{\text{MLP}}(f_{\text{SSA}}(\boldsymbol{e}_{\text{ctr}}) + f_{\text{TSA}}(\boldsymbol{e}_{\text{ctr}})) + \boldsymbol{e}_{\text{ctr}} \tag{19}$$

To simplify the expression, we omit the embedding of diffusion timestep, trajectory timestep and rewards in Equation 16, 17, 18 and 19. In the implementation, the embedding of diffusion timestep and trajectory timestep are added to every input of CCA, SSA and TSA, and the embedding of rewards is added to every input of SSA and TSA.

**Output layer.** We use an MLP layer as the output layer to convert the output of the spatial-temporal encoder layers into the noise we desire to predict.

## C Datasets

### C.1 Detials of Datasets

We apply five real-world traffic flow datasets in three cities of different sizes including Hangzhou and Jinan: (1) **Hangzhou datasets**: the road network contains 16 (4×4) intersections with two traffic flow datasets, including Hangzhou$_1$ and Hangzhou$_2$. (2) **Jinan datasets**: the road network contains 12 (3×4) intersections with three traffic flow datasets, including Jinan$_1$, Jinan$_2$ and Hangzhou$_3$. All these datasets are accessible in https://traffic-signal-control.github.io/.

We train AttendLight [10], Efficient-CoLight [48] and Advanced-CoLight [13] in isolation for each dataset from scratch until convergence. Then we collect all transitions in the replay buffer for each dataset during training. We present the converged performance of three methods in Table 4.

Table 4: Converged performance of methods used to collect offline datasets.

| Methods | Hangzhou$_1$ | Hangzhou$_2$ | Jinan$_1$ | Jinan$_2$ | Jinan$_3$ |
|---|---|---|---|---|---|
| AttendLight | 285.37 | 354.74 | 286.77 | 263.31 | 248.69 |
| Efficient-CoLight | 282.92 | 326.73 | 259.04 | 240.51 | 237.73 |
| Advanced-CoLight | 271.73 | 311.12 | 247.32 | 233.53 | 229.45 |

### C.2 Missing Pattern

In TSC with missing data, missing patterns have a significant impact on the performance of control. In this paper, as shown in Figure 4, we conduct our experiments on random missing and kriging missing: (1) **Random missing**: traffic data collected by sensors for rewards and observations in every intersection is randomly masked with 10%~50% probability. (2) **Kriging missing**: there is always no traffic data collected in certain intersections with 6.25%~25.00% (1-intersection~4-intersection) probability in Hangzhou$_1$ and Hangzhou$_2$, 8.33%~25.00% (1-intersection~3-intersection) probability in Jinan$_1$, Jinan$_2$ and Jinan$_3$, and all neighboring intersections surround missing intersections are observable.

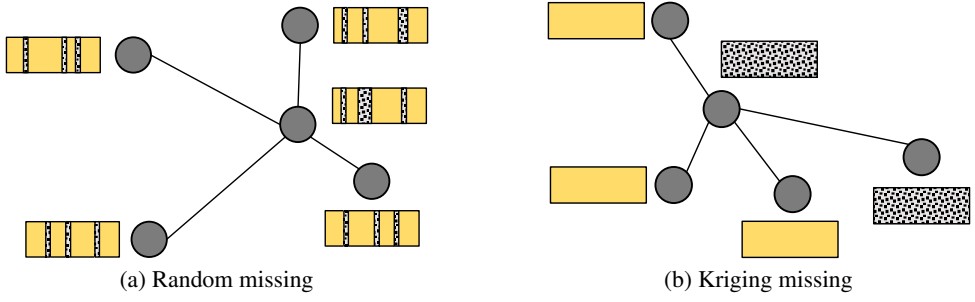

(a) Random missing    (b) Kriging missing

Figure 4: Illustration of random missing and kriging missing pattern. Each node represents an intersection in the road network and blocks with masks are traffic data with missing value.

# D  Baseline Methods

In this section, we make a brief introduction to baseline approaches and the Store-and-forward method (SFM).

**Baseline approaches.**  We adopt six baseline approaches in experiments as follows:

- **BC** is a type of imitation learning, where the agent learns to mimic the behavior of an expert demonstration. The agent is trained on a dataset of state-action pairs from the expert, and the goal is to learn a policy that can replicate the expert's actions given the same states.

- **CQL** is an RL algorithm that aims to learn a conservative Q-function, which provides a lower bound on the true Q-values. This helps to address the issue of overestimation of Q-values, which can lead to poor performance in practice.

- **TD3+BC** combines the TD3 algorithm with behavioral cloning. By incorporating BC into TD3, the agent can leverage expert demonstrations to accelerate learning and improve sample efficiency. TD3+BC offers the benefits of both TD3's stability and BC's ability to learn from expert demonstrations.

- **DT** is a sequence-to-sequence model that casts reinforcement learning as a sequence modeling problem. It takes as input a sequence of past states, actions, and rewards, and it outputs a sequence of future actions that maximize the expected cumulative reward. We build DT based on the code `https://github.com/kzl/decision-transformer/`.

- **Diffuser** is a diffusion-based approach for decision-making. Diffuser focuses on generating sequences of actions that lead to desirable outcomes by iteratively refining these sequences. We build Diffuser based on the code `https://github.com/jannerm/diffuser`.

- **DD** is a diffusion-based approach for decision-making. DD diffuses over state trajectories and planning with an inverse dynamics model. We build DD based on the code `https://github.com/anuragajay/decision-diffuser`.

To avoid non-convergence caused by missing data, we train baselines on datasets without missing data and test them under data-missing scenarios with observations and rewards imputed by SFM.

**Store-and-forward method.**  Since baselines cannot handle the data-missing scenarios, we adopt a rule-based SFM to impute observations and rewards for baselines. It is proved that SFM has more stable performance compared to learning neural networks [14]. In this paper, we model current observation as: $f(\mathcal{V}_{t-1}, k) =$ Concat$(\cup_{l_i} f'(l_i, k))$ and $f'(l_i, k) = \frac{1}{k} \sum_{l_j} o_{t-1}^{l_j}$, where $\mathcal{V}_{t-1}^k$ is the intersection at time $t$, $l_i \in \mathcal{V}_{t-1}^k$ is a lane and $l_j$ is the $k$'s neighboring lane connected by traffic movements. We set $k$ as 12 in this paper.

# E  Proof of Partial Rewards Conditioned Diffusion

To prove that the aim of partial rewards conditioned diffusion is the same as the goal in Equation 3, we assume that the observable part of the trajectory and the missing part of the trajectory are collected by real sensors and virtual sensors separately, and the distribution of traffic data collected by two kinds of sensors are independent. Thus, the distribution in Equation 3 can be factorized as follows,

$$
\begin{aligned}
p(x^0(\boldsymbol{\tau})|\mathrm{y}(\boldsymbol{\tau})) &= p(x^0(\boldsymbol{\tau}_{\mathrm{obs}}), x^0(\boldsymbol{\tau}_{\mathrm{mis}})|y(\boldsymbol{\tau})) \\
&= p(x^0(\boldsymbol{\tau}_{\mathrm{obs}})|y(\boldsymbol{\tau})) \cdot p(x^0(\boldsymbol{\tau}_{\mathrm{mis}})|y(\boldsymbol{\tau})) \\
&= p(x^0(\boldsymbol{\tau}_{\mathrm{obs}})|r(\boldsymbol{\tau}), y'(\boldsymbol{\tau})) \cdot p(x^0(\boldsymbol{\tau}_{\mathrm{mis}})|r(\boldsymbol{\tau}), y'(\boldsymbol{\tau})) \\
&= p(x^0(\boldsymbol{\tau}_{\mathrm{obs}})|r(\boldsymbol{\tau}), y'(\boldsymbol{\tau})) \cdot p(x^0(\boldsymbol{\tau}_{\mathrm{mis}})|y'(\boldsymbol{\tau}))
\end{aligned}
\tag{20}
$$

The distribution without rewards condition $p(x^0(\boldsymbol{\tau})|y'(\boldsymbol{\tau}))$ can be regarded as the marginal distribution of that with rewards condition $p(x^0(\boldsymbol{\tau})|y(\boldsymbol{\tau})) = p(x^0(\boldsymbol{\tau})|r(\boldsymbol{\tau}), y'(\boldsymbol{\tau}))$,

$$
p(x^0(\boldsymbol{\tau})|y'(\boldsymbol{\tau})) = \int p(r(\boldsymbol{\tau}))p(x^0(\boldsymbol{\tau})|r(\boldsymbol{\tau}), y'(\boldsymbol{\tau}))dr(\boldsymbol{\tau})
\tag{21}
$$

In this case, we can adopt the same diffusion model with classifier-free guidance to model $p_\theta(x^0(\boldsymbol{\tau}_{\mathrm{obs}})|r(\boldsymbol{\tau}), y'(\boldsymbol{\tau}))$ and $p_\theta(x^0(\boldsymbol{\tau}_{\mathrm{mis}})|y'(\boldsymbol{\tau}))$.

# F   Additional Experiments Results

## F.1   Performance without Missing Data

We train and test our method on all five datasets and compare our method with all baselines under no data-missing scenarios. DiffLight performs the best on over half of the datasets. Meanwhile, DD demonstrates a better performance than Diffuser, which shows that diffusing only on observations is a better choice in TSC.

Table 5: Overall performance in scenarios without missing data.

| Dataset | BC | CQL | TD3+BC | DT | Diffuser | DD | DiffLight |
|---------|------|------|--------|------|----------|--------|-----------|
| $\mathcal{D}_{HZ}^1$ | 342.26 | 318.42 | 327.19 | 297.46 | 289.64 | 284.79 | **283.92**±0.10 |
| $\mathcal{D}_{HZ}^2$ | 374.9 | 353.04 | 364.8 | 338.33 | 357.34 | 328.63 | **319.79**±5.13 |
| $\mathcal{D}_{JN}^1$ | 315.2 | 298.02 | 304.36 | 289.8 | 270.83 | **255.53** | 268.43±1.35 |
| $\mathcal{D}_{JN}^2$ | 286.66 | 300.64 | 325.53 | 257.59 | 249.74 | 244.11 | **243.56**±0.03 |
| $\mathcal{D}_{JN}^3$ | 296.05 | 288.42 | 281.59 | 247.71 | 241.44 | **241.34** | 242.31±1.39 |

## F.2   Influence of Unobserved Locations

In previous experiments, the unobserved intersections in kriging missing are not adjacent. In this section, we study the influence of unobserved locations. We provide another mask of $\mathcal{D}_{HZ}^1$ with a missing rate of 25% in kriging missing, which contains a missing intersection where all neighboring intersections are missing. The performance of this experiment is shown in Table 6. DiffLight still demonstrates the best performance.

Table 6: Performance in different unobserved locations.

| Dataset | BC | CQL | TD3+BC | DT | Diffuser | DD | DiffLight |
|---------|------|------|--------|------|----------|--------|-----------|
| $\mathcal{D}_{HZ}^1$ w/ neighbors | 354.86 | 328.83 | 341.89 | 381.94 | 328.79 | 836.46 | **302.16**±1.23 |
| $\mathcal{D}_{HZ}^1$ w/o neighbors | 398.77 | 361.11 | 447.79 | 427.62 | 465.61 | 745.85 | **344.02**±8.72 |

## F.3   Limit of Missing Rates

To further explore limits on missing proportions, we conduct experiments on the selected datasets in random missing with missing rates of 70% and 90%. In the experiment, DiffLight remains an acceptable performance at the missing rate of 70%. When the missing rate rises to 90%, the performance of DiffLight drops rapidly, which shows that the limit for the missing rate is around 70%.

Table 7: Limit of missing rates in random missing.

| Dataset | RM(70%) | RM(90%) |
|---------|---------|---------|
| $\mathcal{D}_{HZ}^1$ | 326.29 | 878.31 |
| $\mathcal{D}_{HZ}^2$ | 343.48 | 430.38 |
| $\mathcal{D}_{JN}^1$ | 310.74 | 437.19 |
| $\mathcal{D}_{JN}^2$ | 295.07 | 587.42 |
| $\mathcal{D}_{JN}^3$ | 289.01 | 668.41 |

## F.4   Scalability of DiffLight

To further evaluate the efficacy and validate the performance of our approach, we conduct experiments on the New York dataset, which includes 48 intersections. In the experiment on the New York dataset, DiffLight achieves the best performance in most scenarios, demonstrating its ability to deal with complex traffic scenarios and control traffic signals in a larger-scale traffic network. In contrast, the performance of most baselines drops rapidly, due to the cumulative effect of errors in state imputation and decision-making at more intersections.

Table 8: Scalability of DiffLight in random missing.

| Dataset | Rate | BC | CQL | TD3+BC | DT | Diffuser | DD | DiffLight |
|---------|------|-----|-----|--------|-----|----------|-----|-----------|
| $\mathcal{D}_{NY}$ | 10% | 187.14 | 200.77 | 349.54 | 394.17 | 209.37 | 185.98 | **182.89** |
| | 30% | **226.23** | 254.73 | 540.18 | 605.81 | 241.32 | 229.44 | 244.93 |
| | 50% | 453.90 | 446.29 | 820.19 | 837.97 | 453.97 | 455.07 | **266.82** |

Table 9: Scalability of DiffLight in kriging missing.

| Dataset | Rate | BC | CQL | TD3+BC | DT | Diffuser | DD | DiffLight |
|---------|------|-----|-----|--------|-----|----------|-----|-----------|
| $\mathcal{D}_{NY}$ | 6.25% | 515.40 | 242.15 | 496.41 | 894.76 | 741.99 | 765.64 | **197.22** |
| | 12.50% | 1304.52 | 470.69 | 859.98 | 930.78 | 951.49 | 1213.08 | **315.05** |
| | 18.75% | 1360.71 | 1154.71 | 989.99 | 1197.74 | 1034.02 | 929.25 | **350.66** |
| | 25.00% | 1442.31 | 1089.39 | 1108.67 | 1445.37 | 846.18 | 1393.23 | **454.56** |

## F.5 Additional Ablation Study on Diffusion Communication Mechanism

We further evaluate the effectiveness of DCM with DiffLight w/ DCM and DiffLight w/o DCM. Table 10 shows the comparison of these variants on Hangzhou$_1$ and Jinan$_1$. It should be noted that we adopt another mask of $\mathcal{D}_{HZ}^1$ used in Appendix F.2, which contains a missing intersection where all neighboring intersections are missing. Based on the results, we can find that DiffLight w/ DCM shows better performance in kriging missing and the performance of DiffLight w/ DCM is close to the performance of DiffLight w/o DCM in random missing. It is proven that DCM sharing generated observations among intersections can promote the performance of TSC with missing data effectively.

Table 10: Ablation study on Diffusion Communication Mechanism.

| Dataset | Pattern and Rate | DiffLight w/ DCM | DiffLight w/o DCM |
|---------|------------------|------------------|-------------------|
| $\mathcal{D}_{HZ}^1$ | RM(50%) | 304.71±1.26 | **303.13**±2.23 |
| | KM(25%) | **302.16**±1.23 | 307.14±7.79 |
| $\mathcal{D}_{JN}^1$ | RM(50%) | 290.02±1.26 | **289.16**±3.89 |
| | KM(25%) | **329.67**±16.03 | 351.64±19.53 |
| $\mathcal{D}_{HZ}^1$ | KM(25%) w/o neighbors | **344.02**±8.72 | 351.64±2.46 |

## F.6 Additional Ablation Study on the Inverse Dynamics

We further evaluate the effectiveness of the inverse dynamics (ID) with DiffLight w/ ID and DiffLight w/o ID. For DiffLight w/o inverse dynamics, we remove the inverse dynamics and extend the dimension of the noise model to generate both observations and actions. Table 11 shows the comparison of these variants on Hangzhou$_1$ and Jinan$_1$. Based on the results, we can find that DiffLight w/ inverse dynamics shows better performance in both random missing and kriging missing.

Table 11: Ablation study on the inverse dynamics.

| Dataset | Pattern and Rate | DiffLight w/ ID | DiffLight w/o ID |
|---------|------------------|-----------------|------------------|
| $\mathcal{D}_{HZ}^1$ | RM(50%) | **303.91** | 572.61 |
| | KM(25%) | **301.08** | 386.92 |
| $\mathcal{D}_{JN}^1$ | RM(50%) | **288.01** | 301.21 |
| | KM(25%) | **334.12** | 395.46 |

## F.7 Time Cost

To effectively demonstrate the usability of DiffLight, we conduct experiments to study the relationship between inference speed and performance in Table 12 and 13. We adopt models trained with 100 steps and test them on

different sampling steps. We can see that with the decrease in sampling steps, the performance of DiffLight remains stable. It is proven that DiffLight is able to handle the TSC task in an acceptable time with good performance.

Table 12: Performance of DiffLight on different sampling steps.

| Dataset | Pattern and Rate | 100-step | 50-step | 20-step | 10-step |
|---|---|---|---|---|---|
| $\mathcal{D}_{HZ}^1$ | RM(50%) | 301.62±0.42 | 301.90±2.79 | 300.75±0.39 | 300.95±1.77 |
| $\mathcal{D}_{HZ}^1$ | KM(25%) | 308.51±2.30 | 303.45±1.75 | 303.38±0.07 | 306.77±2.70 |
| $\mathcal{D}_{JN}^1$ | RM(50%) | 288.51±1.45 | 289.45±2.68 | 289.72±0.18 | 287.36±1.29 |
| $\mathcal{D}_{JN}^1$ | KM(25%) | 328.54±0.99 | 337.17±8.71 | 332.31±12.66 | 325.20±6.37 |

Table 13: Inference time cost of DiffLight on different sampling steps.

| Dataset | Pattern and Rate | 100-step | 50-step | 20-step | 10-step |
|---|---|---|---|---|---|
| $\mathcal{D}_{HZ}^1$ | RM(50%) | 450.36±14.32 | 219.81±3.96 | 90.19±2.41 | 45.73±0.16 |
| $\mathcal{D}_{HZ}^1$ | KM(25%) | 442.37±9.56 | 219.87±4.05 | 89.64±3.17 | 46.14±1.00 |
| $\mathcal{D}_{JN}^1$ | RM(50%) | 449.01±7.19 | 218.31±2.69 | 90.23±1.77 | 45.50±0.76 |
| $\mathcal{D}_{JN}^1$ | KM(25%) | 436.30±8.85 | 218.26±2.55 | 88.41±1.47 | 45.92±0.51 |

## G  Discussion on MissLight

To better clarify the core differences between DiffLight and MissLight [14], we compare them from the following two aspects.

**Model training.** MissLight is an online method with a state imputation model and a reward imputation model, which means that interaction with the environment is necessary. In the online setting, if the method is trained in the physical environment, safety problems must be taken into consideration. If the method is trained in a simulated environment, the difference between the physical environment and the simulated environment could affect the performance of the method to some extent when the online method is going to be employed in the physical environment. In contrast, our approach, DiffLight, is an offline method based on the diffusion model. In the offline setting, our method is trained using the collected dataset without interaction with the environment, which avoids the problems mentioned above.

**Model composition.** MissLight is a two-stage method. In the first stage, state imputation and reward imputation models are used to fill in the missing data. In the second stage, the DQN algorithm is employed to complete the training process based on the imputed data. This approach suffers from the problem of error accumulation during the training process. However, our proposed DiffLight model, which incorporates both a diffusion model and an inverse dynamics model, can simultaneously train on missing data and collaboratively achieve traffic signal control with missing data.

To better compare with MissLight, we implement the SDQN-SDQN (model-based) in [14] and replace the DQN algorithm with the CQL algorithm to adapt the offline setting. We replaced the DQN algorithm with different algorithms in the offline setting and imputed the states with the SFM model. Note that all the baselines in Section 4.2 were implemented with reference to the SDQN-SDQN (transferred) method in MissLight. To distinguish the baseline of CQL implemented in Section 4.2, the new baseline is named CQL (model-based). We provide the performance of CQL (model-based) in Table 14 and Table 15.

Table 14: Performance of CQL (model-based) in random missing.

| Dataset | Rate | CQL | CQL (model-based) | DiffLight |
|---|---|---|---|---|
| $\mathcal{D}_{HZ}^1$ | 10% | 363.50 | 376.85 | **285.96** |
| | 30% | 368.53 | 381.92 | **293.10** |
| | 50% | 383.67 | 388.51 | **303.91** |
| $\mathcal{D}_{JN}^1$ | 10% | 299.07 | 303.46 | **273.17** |
| | 30% | 310.80 | 324.15 | **280.32** |
| | 50% | 322.25 | 361.40 | **288.01** |

Table 15: Performance of CQL (model-based) in kriging missing.

| Dataset | Rate | CQL | CQL (model-based) | DiffLight |
|---|---|---|---|---|
| $\mathcal{D}_{HZ}^1$ | 6.25% | 317.69 | 389.66 | **291.80** |
| | 12.50% | 317.94 | 397.13 | **297.18** |
| | 18.75% | 319.18 | 449.80 | **299.96** |
| | 25.00% | 328.83 | 463.25 | **301.08** |
| $\mathcal{D}_{JN}^1$ | 8.33% | 302.35 | 374.20 | **280.83** |
| | 16.67% | 343.16 | 347.88 | **295.53** |
| | 25.00% | 398.66 | 400.55 | **334.12** |

DiffLight achieves competitive performance compared with CQL (transferred) and CQL (model-based). The possible reason why DiffLight has better performance is that CQL (model-based) suffers from error accumulation caused by the reward imputation model while DiffLight can directly make decisions with Partial Rewards Conditioned Diffusion (PRCD).

