# OpenReview forum: "DiffLight: A Partial Rewards Conditioned Diffusion Model for Traffic Signal Control with Missing Data"
_NeurIPS.cc/2024/Conference — NeurIPS 2024 spotlight_

### Official Review · Reviewer_V6Ax · 2024-07-12

**Soundness:** 4
**Presentation:** 3
**Contribution:** 4
**Rating:** 7
**Confidence:** 5

**Summary:**

This paper proposes a conditional diffusion model named DiffLight, which is able to unify traffic data imputation and decision-making for TSC when data is incomplete. Specifically, it proposes a partial reward conditional diffusion method to avoid the negative effects brought by the padded values of missing data. Additionally, a novel spatial-temporal transformer architecture and a diffusion communication mechanism are designed to capture spatiotemporal dependencies between intersections as well as enhance cooperative control among them. Extensive experiments on five datasets demonstrate the effectiveness of the DiffLight. Besides, the source code provided in this paper is accessible and available for use.

**Strengths:**

S1: This paper studies a significant problem of traffic signal control with missing data, which is often overlooked by researchers but is prevalent in real urban traffic scenarios. So, the solutions proposed in this paper will benefit practical use about TSC a lot. Besides, the paper is well-written and easy to follow.
S2: The idea of unifying the problems of traffic data imputation and traffic signal control decision-making in a model is interesting, also wise and reasonable, but has never been studied before. The design of partial reward conditional diffusion (PRCD) with classifier-free guidance is novel, which makes it feasible to model the complex distribution of traffic data, especially in scenarios with missing traffic data. Additionally, the paper provides theoretical proof of PRCD's feasibility to demonstrate how PRCD avoids the negative impacts of padding.
S3: In the noise model of DiffLight, the paper designs the spatial-temporal transformer architecture (STFormer) to model the spatiotemporal dependencies between multiple intersections in traffic signal control. Furthermore, the paper introduces a diffusion communication mechanism, which assists in capturing the spatiotemporal dependencies between multiple intersections by disseminating the observational information generated by the STFormer during the backward process. This mechanism enables effective traffic signal control in difficult scenarios, such as simultaneous data loss at the current intersection and neighboring intersections.

**Weaknesses:**

W1: The authors should further explain how the Diffusion Communication Mechanism (DCM) enables information communication among multiple intersections when data is missing at both the current intersection and neighboring intersections. This part is not clear enough.
W2: This paper only considers traffic signal control under two typical types of missing data patterns. While it is undeniable that random missing and Kriging missing are the most common patterns in real life, future research could explore traffic signal control under a wider variety of missing data patterns, including mixed missing data scenarios.
W3: There are some symbols in the paper that are not correctly defined. For example:
- At line 146, \(\tau\) is defined as the observation trajectory, but at line 152, it states \(\tau_{i}\) represents the neighboring intersection. I think here \(\tau_{i}\) should denote the observation trajectory of the neighboring intersection.
- In the experimental section, at lines 220 and 221, the evaluation metric (Average Travel Time, ATT) is defined. Is there a more objective formula definition and explanation for it?

**Questions:**

Could you provide a specific example to explain the design of the DCM mechanism?

**Limitations:**

See weaknesses.

---

> ### Author Rebuttal · Authors · 2024-08-07
>
> We are delighted that the reviewer found our motivation interesting and reasonable. Thank you for your positive and insightful comments. We respond to each of the points below:
>
> > [W1 & Q1] Explanation of Diffusion Communication Mechanism
>
> Thank you for your comments. We have made a brief introduction in **Section 3.2**. We apologize for our unclear expression and appreciate the opportunity to clarify this. Diffusion Communication Mechanism (DCM) is designed to enhance information communication among multiple intersections. In DCM, the states of neighboring intersection $x^k(\tau^i)$ diffused by diffusion model at diffusion step $k$ would be used to calculate $x^0(\tau^i)$ and then  $x^0(\tau^i)$ would be sent to its current intersections. The agent in the current intersection would take $x^0(\tau^i)$ as the condition to generate the states of the current intersection $x^{k-1}(\tau)$ at diffusion step $k-1$.
>
> > [W2] Missing patterns of data
>
> Thank you for highlighting the complex missing patterns of data in the real world. Undoubtedly, the missing patterns of data in the real world are far more complex than the random and Kriging missing patterns mentioned. We completely agree with this. In future work, we will follow your suggestion to further explore traffic signal control methods under mixed data-missing scenarios.
>
> > [W3] Symbols and metrics
>
> Thank you for your detailed inspection of the symbol definitions. We will carefully correct these incorrect symbol definitions to improve the paper. Additionally, regarding the explanation of average travel time (ATT), it is the most commonly used metric in the field of traffic signal control. This metric represents the average travel time of all vehicles from entering the road network to leaving the intersection, and it effectively reflects the performance of traffic signal control. The calculation formula is as follows:
>
> $$
> \text{ATT}=\frac{1}{N} \sum_{i=1}^{N} \left ( t_{i}^{l}-t_{i}^{e} \right ),
> $$
>
> where $N$ is the total number of vehicles entering the road network, $t_{i}^{e}$ and $t_{i}^{l}$ are the entering time and leaving time for the $i$-th vehicle respectively.

---

> > ### Comment · Reviewer_V6Ax · 2024-08-12
> >
> > Thanks for the detailed responses. I'm generally fine with the results.  Thus, I will keep my score.

---

> > > ### Author Response · Authors · 2024-08-14
> > >
> > > Thank you for taking the time to review and for your thoughtful feedback. We will take your suggestions into consideration in our future work.

---

### Official Review · Reviewer_PkZo · 2024-07-13

**Soundness:** 2
**Presentation:** 3
**Contribution:** 2
**Rating:** 5
**Confidence:** 4

**Summary:**

The paper introduces "DiffLight," a novel approach combining traffic data imputation and decision-making for Traffic Signal Control (TSC) under scenarios with missing data using a diffusion model framework. It employs partial rewards conditioned diffusion and a spatial-temporal transformer architecture to address challenges associated with incomplete data. The proposed model is tested extensively on existing datasets, demonstrating its efficacy in handling different missing data scenarios.

**Strengths:**

1. Originality: The approach of using a conditional diffusion model to simultaneously perform traffic data imputation and decision-making in TSC is straightforward. The integration of partial rewards conditioned diffusion is particularly novel and helps mitigate the impact of data padding issues.

2. Clarity: The paper is clearly written, with a logical structure that carefully explains the methodology and experimental setup, making it accessible and understandable.

**Weaknesses:**

1. There is a significant gap in the comparative analysis. The paper does not benchmark against relevant state-of-the-art methods like "MissLight" and other baseline diffusion models which are crucial for establishing the efficacy of the proposed approach.
2. In MissLight, they used $D_{HZ}$, and $D_{NY}$ but this paper used new dataset, which is not clear why.

**Questions:**

1. Can you explain why MissLight is neglected? If you have offline data, it is easy to employ an offline version of MissLight as well.
2. Why not compare on the existing dataset used in MissLight?

**Limitations:**

The paper discussed limitations.

---

> ### Author Rebuttal · Authors · 2024-08-07
>
> We highly appreciate your high-quality and valuable suggestions. We provide the point-by-point response as follows:
>
> > [W1 & Q1] Comparative analysis
>
> Thank you for emphasizing the gap in the comparative analysis. We apologize for our unclear expression. Actually, all the baselines were implemented with reference to the SDQN-SDQN (transferred) method in MissLight [1]. We replaced the DQN algorithm with different algorithms in the offline setting and imputed the states with the SFM model.
>
> To further evaluate the efficacy of the proposed approach, we employ an offline version of SDQN-SDQN (model-based) in MissLight and conduct experiments on Hangzhou$_1$ and Jinan$_1$. To distinguish the baseline of CQL in Section 4.2, it is named CQL (model-based) while CQL in Section 4.2 is named CQL (transferred).
>
> **Random Missing:**
>
> |Dataset|Rate|CQL (transferred)|CQL (model-based)|DiffLight|
> |-|-|-|-|-|
> |$\mathcal{D}_{\text{HZ}}^1$|10.00%|363.50|376.85|285.96|
> ||30.00%|368.53|381.92|293.10|
> ||50.00%|383.67|388.51|303.91|
> |$\mathcal{D}_{\text{JN}}^1$|10.00%|299.07|303.46|273.17|
> ||30.00%|310.80|324.15|280.32|
> ||50.00%|322.25|361.40|288.01|
>
> **Kriging Missing:**
>
> |Dataset|Rate|CQL (transferred)|CQL (model-based)|DiffLight|
> |-|-|-|-|-|
> |$\mathcal{D}_{\text{HZ}}^1$|6.25%|317.69|389.66|291.80|
> ||12.50%|317.94|397.13|297.18|
> ||18.75%|319.18|449.80|299.96|
> ||25.00%|328.83|463.25|301.08|
> |$\mathcal{D}_{\text{JN}}^1$|8.33%|302.35|374.20|280.83|
> ||16.67%|343.16|347.88|295.53|
> ||25.00%|398.66|400.55|334.12|
>
> In the new experiment, DiffLight achieves competitive performance compared with CQL (transferred) and CQL (model-based). The possible reason why DiffLight has better performance is that CQL (model-based) suffers from error accumulation caused by the reward imputation model while DiffLight can directly make decisions with Partial Rewards Conditioned Diffusion (PRCD).
>
> Besides, Diffuser [2] and Decision Diffuser [3] are the most representative work in the field of diffusion model for RL, which are included as baselines in much literature [9, 10, 11]. Thus, we choose them as our baselines rather than other diffusion-based methods.
>
> > [W2&Q2] Additional datasets
>
> Thank you for highlighting the discussion on the selection of datasets. Actually, Hangzhou, Jinan and New York datasets are all commonly used in recent literature [1, 4, 5, 6, 7, 8]. We conducted extensive experiments on Hangzhou and Jinan datasets with different data-missing scenarios. To further evaluate the efficacy and validate the performance of our approach, we conduct experiments on the **New York** dataset, which includes 48 intersections.
>
> **Random Missing:**
>
> |Dataset|Rate|BC|CQL|TD3+BC|DT|Diffuser|DD|DiffLight|
> |-|-|-|-|-|-|-|-|-|
> |$\mathcal{D}_{\text{NY}}$|10%|187.14|200.77|349.54|394.17|209.37|185.98|182.89|
> ||30%|226.23|254.73|540.18|605.81|241.32|229.44|244.93|
> ||50%|453.90|446.29|820.19|837.97|453.97|455.07|266.82|
>
> **Kriging Missing:**
>
> |Dataset|Rate|BC|CQL|TD3+BC|DT|Diffuser|DD|DiffLight|
> |-|-|-|-|-|-|-|-|-|
> |$\mathcal{D}_{\text{NY}}$|6.25%|515.40|242.15|496.41|894.76|741.99|765.64|197.22|
> ||12.50%|1304.52|470.69|859.98|930.78|951.49|1213.08|315.05|
> ||18.75%|1360.71|1154.71|989.99|1197.74|1034.02|929.25|350.66|
> ||25.00%|1442.31|1089.39|1108.67|1445.37|846.18|1393.23|454.56|
>
> In the new experiment on the New York dataset, DiffLight achieves the best performance in most of scenarios, demonstrating its ability to deal with complex traffic scenarios and control traffic signals in a larger scale traffic network. In contrast, the performance of most baselines drops rapidly, due to the cumulative effect of errors in state imputation and decision-making at more intersections.
>
> Reference:
>
> [1] Mei, Hao, et al. "Reinforcement learning approaches for traffic signal control under missing data." Proceedings of the Thirty-Second International Joint Conference on Artificial Intelligence. 2023.
>
> [2] Janner, Michael, et al. "Planning with Diffusion for Flexible Behavior Synthesis." *International Conference on Machine Learning*. PMLR, 2022.
>
> [3] Ajay, Anurag, et al. "Is Conditional Generative Modeling all you need for Decision Making?." *The Eleventh International Conference on Learning Representations*.
>
> [4] Ye, Yutong, et al. "InitLight: initial model generation for traffic signal control using adversarial inverse reinforcement learning." Proceedings of the Thirty-Second International Joint Conference on Artificial Intelligence. 2023.
>
> [5] Wei, Hua, et al. "Colight: Learning network-level cooperation for traffic signal control." Proceedings of the 28th ACM international conference on information and knowledge management. 2019.
>
> [6] Zang, Xinshi, et al. "Metalight: Value-based meta-reinforcement learning for traffic signal control." Proceedings of the AAAI conference on artificial intelligence. Vol. 34. No. 01. 2020.
>
> [7] Yu, Zhengxu, et al. "MaCAR: Urban traffic light control via active multi-agent communication and action rectification." Proceedings of the Twenty-Ninth International Conference on International Joint Conferences on Artificial Intelligence. 2021.
>
> [8] Zhang, Liang, et al. "Expression might be enough: representing pressure and demand for reinforcement learning based traffic signal control." International Conference on Machine Learning. PMLR, 2022.
>
> [9] Li, Wenhao, et al. "Hierarchical diffusion for offline decision making." *International Conference on Machine Learning*. PMLR, 2023.
>
> [10] Venkatraman, Siddarth, et al. "Reasoning with Latent Diffusion in Offline Reinforcement Learning." *The Twelfth International Conference on Learning Representations*.
>
> [11] Zhu, Zhengbang, et al. "Madiff: Offline multi-agent learning with diffusion models." *arXiv preprint arXiv:2305.17330* (2023).

---

> > ### Comment · Reviewer_PkZo · 2024-08-11
> >
> > Thanks for the clarification on the comparisons to existing baselines. These new experimental results help us understand what actually helps under the missing data setting. From what I see, the offline RL method (CQL) definitely helps. I'm raising my score +1 and suggest you add the above results later in this paper.
> >
> > As for the ablation study, it seems like we still miss some ablation on the Inverse Dynamics part. I understand it was originally part of the Decision Diffuser but I'm still curious to see its performance.

---

> > > ### Author Response · Authors · 2024-08-14
> > >
> > > We are truly grateful to the reviewer for the invaluable insights and detailed feedback. We will add the content of our discussion to the subsequent version.
> > >
> > > To evaluate the performance of the inverse dynamics, we remove the inverse dynamics and extend the dimension of the noise model to generate both observations and actions.
> > >
> > > **Random Missing:**
> > >
> > > | Dataset                     | Rate   | w/o inverse dynamics | w/ inverse dynamics |
> > > | --------------------------- | ------ | -------------------- | ------------------- |
> > > | $\mathcal{D}_{\text{HZ}}^1$ | 50.00% | 572.61               | 303.91              |
> > > | $\mathcal{D}_{\text{JN}}^1$ | 50.00% | 301.21               | 288.01              |
> > >
> > > **Kriging Missing:**
> > >
> > > | Dataset                     | Rate   | w/o inverse dynamics | w/ inverse dynamics |
> > > | --------------------------- | ------ | -------------------- | ------------------- |
> > > | $\mathcal{D}_{\text{HZ}}^1$ | 25.00% | 386.92               | 301.08              |
> > > | $\mathcal{D}_{\text{JN}}^1$ | 25.00% | 395.46               | 334.12              |
> > >
> > > In the ablation experiment of the inverse dynamics model, our approach with the inverse dynamics model achieves better performance.

---

### Official Review · Reviewer_3ABW · 2024-07-13

**Soundness:** 3
**Presentation:** 2
**Contribution:** 3
**Rating:** 6
**Confidence:** 4

**Summary:**

The paper proposes a conditional diffusion framework to address the traffic signal control (TSC) problem under conditions of missing data. Utilizing Partial Rewards Conditioned Diffusion (PRCD) with classifier-free guidance, for both data imputation and decision making. The authors employ the DDIM sampling method and a spatial-temporal transformer noise predictor model to reconstruct the observation trajectory. The conditioning information is derived from partially observable rewards. A Diffusion Communication Mechanism (DCM) is introduced to exchange observation information between neighboring intersections. Finally, an inverse dynamic model is used to generate actions. Extensive experiments are conducted on datasets from multiple cities, with different missing patterns. The performance of the proposed approach is compared against multiple baseline methods.

**Strengths:**

1. Traffic signal control is a significant practical issue, and data missing is common in real-world applications. Solving this problem has substantial practical implications.

2. The use of diffusion models to unify data imputation and 10 decision-making tasks in TSC is novel.

**Weaknesses:**

1. The selected datasets typically have few intersections and lack large-scale (e.g. thousands of intersections) datasets to validate scalability and cooperative performance.

2. The comparison benchmarks and metrics are limited, lacking some methods, such as RL-based methods and GNN-based methods. Additionally, there is a lack of comparison for certain metrics, such as queue length and throughput.

3. This paper contains many notations, formulas, and conclusions that are directly quoted without specific contextual explanations, making it confusing when reading for the first look.

**Questions:**

What boundary conditions ensure the effectiveness of this method?
Apart from assumptions such as missing data being independent of existing data, are there constraints such as limits on missing proportions and conflicts with neighboring traffic?

**Limitations:**

The authors discussed the limitations of the work in the Appendix. It is suggested that they put this part in the main body of the paper.

---

> ### Author Rebuttal · Authors · 2024-08-07
>
> We thank the reviewer for their thoughtful and insightful comments. We respond to each of the points as follows:
>
> > [W1] Selection for datasets
>
> Thank you for your comment. We appreciate the opportunity to elaborate on this. We would like to point out that DiffLight is designed to address the traffic signal control problem under data-missing scenarios, instead of focusing on scalability. Meanwhile, the selected datasets are commonly used in most of the literature [1, 2, 3, 4, 5, 6]. Besides, due to the limitation of computing resources and time, it is difficult to conduct experiments on large-scale datasets with thousands of intersections.
>
> However, there is no denying that scalability and cooperative performance play an essential role in applications. To further validate the scalability and cooperative performance of our approach, we conduct experiments on a larger dataset **New York**, which includes 48 intersections.
>
> **Random Missing:**
>
> |Dataset|Rate|BC|CQL|TD3+BC|DT|Diffuser|DD|DiffLight|
> |-|-|-|-|-|-|-|-|-|
> |$\mathcal{D}_{\text{NY}}$|10%|187.14|200.77|349.54|394.17|209.37|185.98|182.89|
> ||30%|226.23|254.73|540.18|605.81|241.32|229.44|244.93|
> ||50%|453.90|446.29|820.19|837.97|453.97|455.07|266.82|
>
> **Kriging Missing:**
>
> |Dataset|Rate|BC|CQL|TD3+BC|DT|Diffuser|DD|DiffLight|
> |-|-|-|-|-|-|-|-|-|
> |$\mathcal{D}_{\text{NY}}$|6.25%|515.40|242.15|496.41|894.76|741.99|765.64|197.22|
> ||12.50%|1304.52|470.69|859.98|930.78|951.49|1213.08|315.05|
> ||18.75%|1360.71|1154.71|989.99|1197.74|1034.02|929.25|350.66|
> ||25.00%|1442.31|1089.39|1108.67|1445.37|846.18|1393.23|454.56|
>
> In the new experiment on the New York dataset, DiffLight achieves the best performance in most of scenarios, demonstrating its ability to deal with complex traffic scenarios and control traffic signals in a larger scale traffic network. In contrast, the performance of most baselines drops rapidly, due to the cumulative effect of errors in state imputation and decision-making at more intersections.
>
> > [W2] Benchmarks and metrics
>
> Thank you for pointing out the lack of benchmarks and metrics. We apologize for our oversight and appreciate the opportunity to make a further explanation.
>
> 1. **Concerning comparison benchmarks**, we have included RL-based methods in experiments, including CQL and TD3+BC. To our knowledge, there is no GNN-based method proposed for the traffic singal control problem in the offline setting. Therefore, we choose CoLight [3], a GAT-based method widely used in the TSC task, as an additional baseline, to replace the DQN algorithm with the CQL algorithm, and conduct experiments on Hangzhou$_1$ and Jinan$_1$. However, due to the gap between offline RL and online RL, CoLight in the offline setting does not converge stably.
> 2. **Concerning metrics**, average travel time, queue length and throughput do reflect the performance of methods in different aspects. Whereas average travel time is widely used in most of the literature [1, 2, 3, 4, 6] and is a more important metric for improving traffic efficiency. We apologize for our oversight and we will take queue length and throughput into consideration in future work in order to evaluate method performance more comprehensively.
>
> > [W3] Lack of explanations for notations, formulas and conclusions
>
> Thank you for your feedback. We will revisit our paper carefully and incorporate specific contextual explanations for these notations, formulas and conclusions.
>
> > [Q1] Boundary conditions of the method
>
> Thank you for highlighting the boundary conditions of the method. These are indeed important for the application of our approach and I'm pleased to make the following discussion:
>
> 1. **Limits on missing proportions:** To further explore limits on missing proportions, we conduct experiments on the selected datasets in random missing with missing rates of 70% and 90%.
>
> |Dataset|Rate|DiffLight|
> |-|-|-|
> |$\mathcal{D}_{\text{HZ}}^1$|70.00%|326.29|
> ||90.00%|878.31|
> |$\mathcal{D}_{\text{HZ}}^2$|70.00%|343.48|
> ||90.00%|430.38|
> |$\mathcal{D}_{\text{JN}}^1$|70.00%|310.74|
> ||90.00%|437.19|
> |$\mathcal{D}_{\text{JN}}^2$|70.00%|295.07|
> ||90.00%|587.42|
> |$\mathcal{D}_{\text{JN}}^3$|70.00%|289.01|
> ||90.00%|668.41|
>
> In the new experiment, DiffLight remains acceptable performance at the missing rate of 70%. When the missing rate rises to 90%, the performance of DiffLight drops rapidly, which shows that the limit for the missing rate is around 70%.
>
> 2. **Conflicts with neighboring traffic:** In the traffic network, all agents in the intersections make decisions at the same time with observations of the current intersection and neighboring intersections as input. Thus, there is no conflict in the order of decision-making. However, under data-missing scenarios, especially when the current intersection and neighboring intersections are all unobservable, agents' input could be unavailable, leading to conflict which could reduce the performance of the method. We design Diffusion Communication Mechanism (DCM) to alleviate this problem and conduct experiments in **Appendix G.3**.
>
> Reference:
>
> [1] Mei, Hao, et al. "Reinforcement learning approaches for traffic signal control under missing data." IJCAI. 2023.
>
> [2] Ye, Yutong, et al. "InitLight: initial model generation for traffic signal control using adversarial inverse reinforcement learning." IJCAI. 2023.
>
> [3] Wei, Hua, et al. "Colight: Learning network-level cooperation for traffic signal control." CIKM. 2019.
>
> [4] Zang, Xinshi, et al. "Metalight: Value-based meta-reinforcement learning for traffic signal control." AAAI. 2020.
>
> [5] Yu, Zhengxu, et al. "MaCAR: Urban traffic light control via active multi-agent communication and action rectification." IJCAI. 2021.
>
> [6] Zhang, Liang, et al. "Expression might be enough: representing pressure and demand for reinforcement learning based traffic signal control." ICML. PMLR, 2022.

---

> > ### Comment · Reviewer_3ABW · 2024-08-12
> >
> > Thank you for the responses. I'm generally fine with the results. One question about the new results: why are the missing rates for the two experiments (random vs Kriging) totally different?
> >
> > I will keep my score.

---

> > > ### Author Response · Authors · 2024-08-14
> > >
> > > We sincerely thank the reviewer for the feedback and appreciate the opportunity to clarify the question.
> > >
> > > We would like to point out that challenges posed by random missing and Kriging missing are inherently different. In Kriging missing, data collected at unobservable intersections is absent all the time. Therefore, decisions made by agents at unobservable intersections can only be made based on observations of neighboring intersections. In contrast, decisions can be made based on observations of current and neighboring intersections in random missing. Thus, compared with random missing scenarios, Kriging missing scenarios are more complex. Moreover, experiments in the literature [1, 2, 3, 4] on Kriging missing are often conducted with missing rates ranging from 0% to 25% while experiments in the literature [5, 6, 7] on random missing are often conducted with missing rates ranging from 0% to 50% or more. Furthermore, in practice, the performance of many baselines and the proposed method drops dramatically in Kriging missing with a missing rate of 25%, making it challenging to conduct further experiments with higher missing rates.
> > >
> > > Reference:
> > >
> > > [1] Wu, Yuankai, et al. "Inductive graph neural networks for spatiotemporal kriging." *Proceedings of the AAAI Conference on Artificial Intelligence*. Vol. 35. No. 5. 2021.
> > >
> > > [2] Zheng, Chuanpan, et al. "Increase: Inductive graph representation learning for spatio-temporal kriging." *Proceedings of the ACM Web Conference 2023*. 2023.
> > >
> > > [3] Mei, Hao, et al. "Uncertainty-aware Traffic Prediction under Missing Data." *2023 IEEE International Conference on Data Mining (ICDM)*. IEEE, 2023.
> > >
> > > [4] Mei, Hao, et al. "Reinforcement learning approaches for traffic signal control under missing data." Proceedings of the Thirty-Second International Joint Conference on Artificial Intelligence. 2023.
> > >
> > > [5] Tashiro, Yusuke, et al. "Csdi: Conditional score-based diffusion models for probabilistic time series imputation." *Advances in Neural Information Processing Systems* 34 (2021): 24804-24816.
> > >
> > > [6] Liu, Mingzhe, et al. "Pristi: A conditional diffusion framework for spatiotemporal imputation." *2023 IEEE 39th International Conference on Data Engineering (ICDE)*. IEEE, 2023.
> > >
> > > [7] Kollovieh, Marcel, et al. "Predict, refine, synthesize: Self-guiding diffusion models for probabilistic time series forecasting." *Advances in Neural Information Processing Systems* 36 (2023).

---

### Official Review · Reviewer_qAN4 · 2024-07-13

**Soundness:** 4
**Presentation:** 3
**Contribution:** 3
**Rating:** 7
**Confidence:** 3

**Summary:**

This paper focuses on traffic signal control under the condition of missing data, presenting DiffLight, a conditional diffusion model that integrates traffic data imputation and decision-making tasks. It introduces a partial rewards conditioned diffusion method to handle missing rewards (PRCD), employs a spatial-temporal transformer for noise modeling to capture intersection dependencies (STFormer), and proposes a diffusion communication mechanism for coordinated intersection control (DCM). Experiments demonstrate the effectiveness of DiffLight over various baselines. Further, the three components of DiffLight are studied and shown to be individually effective.

**Strengths:**

The paper is overall well written and presented clearly. A rigorous study of the suggested method is performed experimentally, confirming the contribution of each part of the DiffLight framework. Furthermore experiments showing the method can generalize well between differing levels of missing data are useful as this is likely the case in a real-world scenario.

This work may be relevant to the broader RL community as it uniquely handles missing data in diffusion based RL.

**Weaknesses:**

[W1] A direct discussion or comparison to [14] would be appropriate as it addresses a substantially similar problem. While it is primarily an online method, it does have an offline stage drawing the work closer to the proposed method. However, by comparing the presented results in this paper and [14] it seems unlikely [14] would perform better.

**Questions:**

**Minor comments**:

Line 50 "In addition, traffic data imputation for TSC with missing data is necessary." - somewhat repetitive

Line 121 "Diffuer"

Line 196 "in both current intersection" - in both "the" current intersection

Section 3.2 might be revised for clarity.

Consider including a requirements or setup file declaring the dependency versions used for the experiments in the paper. I had no issue reproducing some of the results, however dependencies may change substantially in the future.

**Limitations:**

Any potential negative societal impacts have been addressed.

---

> ### Author Rebuttal · Authors · 2024-08-07
>
> We appreciate the reviewer recognizing the importance of our work and thank you for the detailed comments. Please find the point-by-point responses to the reviewer's comments below.
>
> > [W1] Discussion on MissLight
>
> Thank you for highlighting the importance of a direct discussion between DiffLight and MissLight [1]. We apologize for our unclear expression and appreciate the opportunity to have a further discussion. We have made a brief discussion in the part of the Introduction in lines 40-44. To better clarify the core differences between our approach and MissLight, we compare them from the following two aspects:
>
> 1. **Model training:** MissLight is an online method with a state imputation model and a reward imputation model, which means that interaction with the environment is necessary. In the online setting, if the method is trained in the physical environment, safety problems must be taken into consideration. If the method is trained in a simulated environment, the difference between the physical environment and the simulated environment could affect the performance of the method to some extent when the online method is going to be employed in the physical environment. In contrast, our approach, DiffLight, is an offline method based on the diffusion model. In the offline setting, our method is trained using the collected dataset without interaction with the environment, which avoids the problems mentioned above.
> 2. **Model composition:** MissLight is a two-stage method. In the first stage, state imputation and reward imputation models are used to fill in the missing data. In the second stage, the DQN algorithm is employed to complete the training process based on the imputed data. This approach suffers from the problem of error accumulation during the training process. However, our proposed DiffLight model, which incorporates both a diffusion model and an inverse dynamics model, can simultaneously train on missing data and collaboratively achieve traffic signal control with missing data.
>
> Additionally, it should be noted that MissLight as an online method can't be compared with DiffLight directly. To better compare with MissLight, we implement the SDQN-SDQN (model-based) in MissLight and replace the DQN algorithm with the CQL algorithm in order to adapt the offline setting. Note that all the baselines were implemented with reference to the SDQN-SDQN (transferred) method in MissLight. We replaced the DQN algorithm with different algorithms in the offline setting and imputed the states with the SFM model. To distinguish the baseline of CQL implemented in Section 4.2, the new baseline is named CQL (model-based) while CQL in Section 4.2 is named CQL (transferred).
>
> **Random Missing:**
>
> |Dataset|Rate|CQL (transferred)|CQL (model-based)|DiffLight|
> |-|-|-|-|-|
> |$\mathcal{D}_{\text{HZ}}^1$|10.00%|363.50|376.85|285.96|
> ||30.00%|368.53|381.92|293.10|
> ||50.00%|383.67|388.51|303.91|
> |$\mathcal{D}_{\text{JN}}^1$|10.00%|299.07|303.46|273.17|
> ||30.00%|310.80|324.15|280.32|
> ||50.00%|322.25|361.40|288.01|
>
> **Kriging Missing:**
>
> |Dataset|Rate|CQL (transferred)|CQL (model-based)|DiffLight|
> |-|-|-|-|-|
> |$\mathcal{D}_{\text{HZ}}^1$|6.25%|317.69|389.66|291.80|
> ||12.50%|317.94|397.13|297.18|
> ||18.75%|319.18|449.80|299.96|
> ||25.00%|328.83|463.25|301.08|
> |$\mathcal{D}_{\text{JN}}^1$|8.33%|302.35|374.20|280.83|
> ||16.67%|343.16|347.88|295.53|
> ||25.00%|398.66|400.55|334.12|
>
> DiffLight achieves competitive performance compared with CQL (transferred) and CQL (model-based). The possible reason why DiffLight has better performance is that CQL (model-based) suffers from error accumulation caused by the reward imputation model while DiffLight can directly make decisions with Partial Rewards Conditioned Diffusion (PRCD).
>
> >  [Q1] Word and grammatical errors
>
> Thank you for your feedback. We will correct the word and grammatical errors you mentioned and check for other possible errors.
>
> > [Q2] Code reproducibility
>
> Thank you for your suggestion. We update a requirement into our code in the anonymous link. To enhance the reproducibility, we will include a readme file in the code. We apologize for our oversight of the readability and reproducibility.
>
> Reference:
>
> [1] Mei, Hao, et al. "Reinforcement learning approaches for traffic signal control under missing data." Proceedings of the Thirty-Second International Joint Conference on Artificial Intelligence. 2023.

---

> > ### Comment · Reviewer_qAN4 · 2024-08-12
> >
> > Thank you for addressing my concerns! I will keep my score as is.

---

> > > ### Author Response · Authors · 2024-08-14
> > >
> > > Thank you for your response. Your suggestion is invaluable and we will follow your guidance to incorporate specific contextual explanations in the subsequent version.

---

### Public Comment · ~Junxian_Li1 · 2025-08-03
**Nice work to make further efforts on Missing data scenario!**

Dear authors,
Hello! I'm the second author of MissLight.  I'm glad to see that you discovered new methods for a better training strategy on the task we proposed. In fact, we find that the convergence of simple DQN may sometimes fail. Thank you for your contribution!

Best regards,
Junxian Li

---

> ### Public Comment · ~Hanyang_Chen1 · 2025-08-05
>
> Dear Junxian,
>
> Thank you for your comment! MissLight really inspires us a lot. Hope to have further discussions with you in the future!
>
> Best regards,
>
> Hanyang Chen

---

### Decision · Program_Chairs · 2024-09-25

**Decision:**

Accept (spotlight)

**Comment:**

This paper tackles the practical application of traffic signal control under missing data, proposes a novel diffusion-based model architecture, and performs a sound evaluation that shows the proposed approach outperforms various relevant baselines and provides insights through ablation. Presentation is generally clear (though some typos and grammar errors must be fixed for the camera ready version). Moreover, as pointed by reviewers, the approach can be useful for other applications, making it relevant to the broader RL community.

The new experimental results (comparison with MissLight, results on larger-scale NY dataset, boundary conditions on missing rates...) and explanations (e.g., why kriging is more challenging) provided during the rebuttal period make the paper clearer and sounder and should be included in the camera ready version.